# Chiral symmetry breaking yields the $I$-Au$_{60}$ perfect golden shell of singular rigidity

S.-M. Mullins [1], H.-Ch. Weissker [2,3], R. Sinha-Roy [2,3], J.J. Pelayo[4], I.L. Garzón[5], R.L. Whetten[1] &
X. López-Lozano[1]

The combination of profound chirality and high symmetry on the nm-scale is unusual and would open exciting avenues, both fundamental and applied. Here we show how the unique electronic structure and bonding of quasi-2D gold makes this possible. We report a chiral symmetry breaking, i.e., the spontaneous formation of a chiral-icosahedral shell ($I-$Au$_{60}$) from achiral ($I_h$) precursor forms, accompanied by a contraction in the Au–Au bonding and hence the radius of this perfect golden sphere, in which all 60 sites are chemically equivalent. This structure, which resembles the most complex of semi-regular (Archimedean) polyhedra ($3^4.5^*$), may be viewed as an optimal solution to the topological problem: how to close a 60-vertex 2D (triangular) net in 3D. The singular rigidity of the $I-$Au$_{60}$ manifests in uniquely discrete structural, vibrational, electronic, and optical signatures, which we report herein as a guide to its experimental detection and ultimately its isolation in material forms.

---

[1] Department of Physics and Astronomy, The University of Texas at San Antonio, One UTSA Circle, San Antonio, TX 78249-0697, USA. [2] Aix Marseille University, CNRS, CINaM UMR 7325, 13288 Marseille, France. [3] European Theoretical Spectroscopy Facility, https://www.etsf.eu. [4] Escuela Superior de Apan, Universidad Autónoma del Estado de Hidalgo, Chimalpa Tlalayote, Municipio de Apan, 43920 Hidalgo, Mexico. [5] Instituto de Física, Universidad Nacional Autónoma de México, Apartado Postal 20-364, 01000 México, D.F., Mexico. Correspondence and requests for materials should be addressed to X.L.-L. (email: Xochitl.LopezLozano@utsa.edu)

The unique properties of elemental gold ($Z = 79$, Au) derive from the extreme relativistic contraction of its ionic core $[Xe]\ 5d^{10}$ orbitals. Among these are a propensity toward planarity (two-dimensional (2D) bonding), and a high electro-negativity (2.54) exceeding that of any other metallic or semi-metallic element. Low-dimensionality gold nanostructures have attracted great interest in recent years, especially given their relevance to the activity of supported gold as low-temperature catalysts[1]. The ubiquity of 2D (planar close-packed 'rafts') and quasi-2D (hollow shells or capsules) clusters is well established; they are competitive with compact three-dimensional (3D) structures for the negatively charged forms. Physically, this unusual favoritism for reduced dimensionality structures (those with local planarity) is explained by the profound local hybridization ($|5d_{z^2}\rangle \pm |6s\rangle$) on each Au atom, where $z$ defines a local axis normal to the plane. Generally, with increasing number of atoms, the closed quasi-2D hollow shells (sometimes called 'cages') become more important, as the curvature energy decreases. Globular forms further minimize the strain energy, and among these the icosahedral ($I_h$) shells have the highest symmetry. Icosahedral shells investigated previously include the celebrated $I_h$−Au$_{32}$[2] (it is competitive against all compact neutral forms), as well as higher $I_h$ structures (42, 92, 122 sites)[3]. Karttunen et al.[4] and Johansson et al.[5] explored a shell of broken ($I$) symmetry with 72 atoms (discussed separately below); the charge-neutral clusters $I_h$−Au$_{32}$ and $I$−Au$_{72}$ structures are 'doubly magic' for supporting closed electronic as well as geometric shells.

Chiral-icosahedral ($I$) symmetry, also known as chiral-rotational symmetry, is a reduced (or broken) form of icosahedral ($I_h$) symmetry in which no reflection plane (or center of inversion) remains. It is so rarely considered relevant to finite (molecular) systems as to be dismissed as non-existent[6]. One reason for this neglect is that whereas full $I_h$-symmetry is manifested by as few as 30, 20, or even 12 equivalent sites (atoms), $I$-symmetry requires at minimum 60-fold equivalence to be distinguished. However, it is an accepted feature of the capsids (protein shell) of many icosahedral viruses on a much longer scale (~5 nm). These involve thousands of atoms in each unit versus molecular or solid-state structures that are <1 nm and one or a few atoms in the repeated unit[7]. One clear example with "perfect" icosahedral symmetry is the satellite tobacco necrosis virus. The 60 identical protein subunits are arranged in an icosahedral shell around a single RNA molecule. Many such icosahedral capsids actually exhibit chirality[8].

In the inorganic world, the principal claim to $I$-symmetry structures has come from experiments on metal-atom overlayers on a fullerene-C$_{60}$ molecule. Martin et al.[9] proposed that the enhanced stability (abundance), attained at certain counts—i.e., 32; 32+72*; 32+72*+132*—of alkaline-earth (Ca, Sr)-atoms, should be explained not only by closing of icosahedral ($I_h$) shells, which reflects the symmetry of the $I_h$−C$_{60}$ 'template', but that the outer layers may attain a more compact form by a chiral symmetry breaking (XSB). Ultimately, Martin related these inorganic phenomena directly to those of the viral capsids[10]. Subsequently, Bochicchio and Ferrando[11] have explored $I$-symmetry reconstructions of the surface layer of bimetallic clusters with icosahedral shell cores, the smallest of which has a 72*-site outer layer. Note that none of these cases is a free-standing shell; they are all supported by an inner metal layer comprising 32 or 42 sites.

Inspired by the high stability and the size of the icosahedral 144-atom noble-metal cluster compound[12–16], we explored the chirality of its constitutive shells[17]. During our study, we extracted and analyzed the $I_h$−Au$_{60}$ shell by a structural optimization as indicated in the Supplementary Methods. We expected the collapse of the $I_h$−Au$_{60}$ structure into a compact cluster, similar or close to the 3D $C_1$−Au$_{60}$ discussed in ref. [18].

However, the extracted shell contracts spontaneously and then transforms coherently into a final structure $I$−Au$_{60}$ of remarkable perfection, in which all 60 atoms are in symmetry-equivalent sites and five (5) unusually short interatomic distances (bonds). To our knowledge, the $I$−Au$_{60}$ structure is a unique instance of stability wherein all (60) atoms are equivalent, without the presence of any other supporting elements. This exceptional symmetry characteristic has special consequences that we explore hereby.

## Results

**An $I$−Au$_{60}$ shell of surprising stability.** The structure of Au$_{144}$(SR)$_{60}$ (Fig. 1a) comprises four concentric shells of atoms: the Inner Core of two shells of (12) + (30 + 12) sites that have no coordination to ligands; the Grand Core (114 Au atoms) consists of the Inner Core plus 60 Au surface atoms, hereinafter referred to as the $I_h$−Au$_{60}$ shell (Fig. 1b), each singly coordinated to thiolates (RS-); the final (4th) shell is conformed by 30 staple-motif units (RS-Au(I)-SR). See Fig. 1a and Supplementary Figure 1. The initial geometry of the $I_h$−Au$_{60}$ shell approximates a rhombicosidodecahedron (3.4.5.4), an Archimedean solid, Fig. 1c, d. It has 20 triangular faces (red), 30 square faces (white), 12 pentagonal faces (blue), 60 vertices, and 120 edges. The coordination number for the $I_h$−Au$_{60}$ is four with two initial Au−Au distances: two bond pentagons of 3.32 Å and two squares of 2.96 Å, see Fig. 1c. In order to compare the properties of the 60-atom Au shell with similar cluster sizes, we considered the $I$−Au$_{72}$ nanoshell and the 3D $C_1$−Au$_{60}$, see Fig. 1g, h. Obtained as a compound of concentric polyhedra, the $I$−Au$_{72}$ shell is of particular importance, not only for being the smallest known chiral molecule that exhibits spherical aromaticity[19,20], but for its potential as an enantioselective catalyst[4].

Figure 1b, c shows the initial $I_h$−Au$_{60}$ shell. The relaxed lowest-energy structure of the $I$−Au$_{60}$ (e), the $I$−Au$_{72}$ (g) shells, as well as the 3D $C_1$−Au$_{60}$ (h) are also shown. Coordinate sets are available in the Supplementary Data 1–4. Figure 2a shows the radial distances of the 60-atom Au shell prior [$I_h$−Au$_{60}$ (red)] and after [$I$−Au$_{60}$ (black)] relaxation. For comparison purposes, the radial distances of the $I$−Au$_{72}$ (green) shell have been included. With regard to the structural stability, the 3D $C_1$−Au$_{60}$ cluster is the most stable of all. The difference in total energy between the $C_1$−Au$_{60}$ and $I$−Au$_{60}$ is 12.75 eV, or 0.21 eV/atom, which reduces to 0.03 eV/atom for $I$−Au$_{72}$. The formation energy of $C_1$−Au$_{60}$ is 0.647 eV, for $I$−Au$_{72}$ 0.679 eV, while for $I$−Au$_{60}$ 0.860 eV.

As mentioned above, the $I_h$−Au$_{60}$ transforms spontaneously into another structure resembling the 60-vertex Archimedean solid geometry, namely, the snub dodecahedron (3$^4$.5*), see Supplementary Figures 2, 3 and Supplementary Movie 1 for animation. This interesting structural transformation consists first of the uniform contraction of the entire initial structure followed by a rotation of the pentagonal faces by ~18°[9]. Of its 92 faces, 12 are pentagons (blue) and 80 triangles (red, white), see Fig. 1f. It also has 150 edges or Au–Au 'bonds'. The coordination number increases from four (4) to five (5). Figure 1e shows the ball-and-stick atomic model of the optimized $I$−Au$_{60}$ shell. This transformation is geometrically similar to that observed in AgCu chiral nanoparticles[11]. The transition is accompanied by a contraction of the bonds: two pentagonal edges with a length of 2.70 Å, one triangular edge of 2.74 Å, and two triangular edges of 2.79 Å. In contrast with the regular polyhedra that has 80 equilateral triangles, here 20 are equilateral (red) and 60 isosceles (white), see Fig. 1f. The contracted interatomic distances in the $I$−Au$_{60}$ are reflected in the radial distributions shown in Fig. 2a. Figure 2b shows the interatomic distances of the optimized $I$−Au$_{60}$ (black) and $I$−Au$_{72}$ (green) shells. The 60 shortest bonds

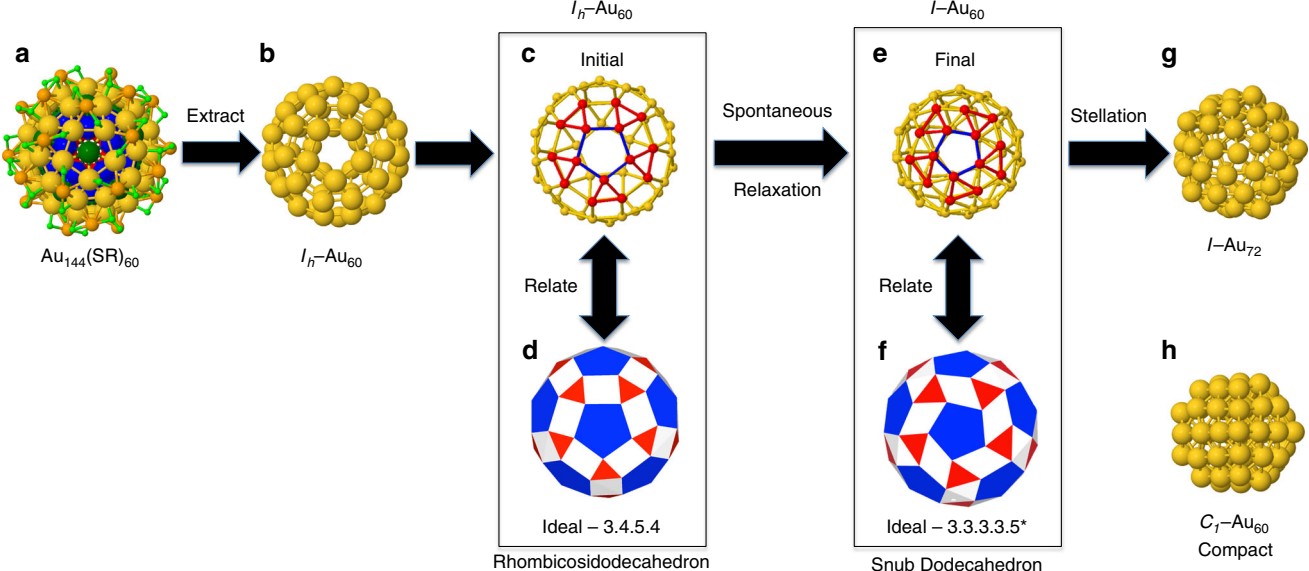

**Fig. 1** Spontaneous formation of the $I-Au_{60}$ shell. **a** Atomic structure of $Au_{144}(SR)_{60}$ from reference[13], **b** the $I_h-Au_{60}$ constitutive shell derived from $Au_{144}(SR)_{60}$, **c** and **e** show the ball-and-stick atomic models of the $Au_{60}$ shell before ($I_h-Au_{60}$) and after ($I-Au_{60}$) atomic relaxation. Some Au atoms located in two opposite pentagonal facets are displayed in different colors to show the structural transition. Note the significant contraction of the 60-atom Au shell. **d** Rhombicosidodecahedron and **f** Snub dodecahedron Archimedean polyhedra, **g** the $I-Au_{72}$ nanoshell, and **h** compact $C_1-Au_{60}$ nanoparticle

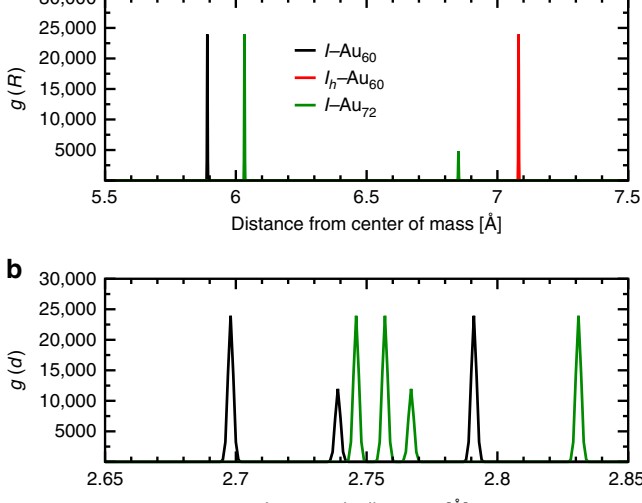

**Fig. 2** Atomic structure comparison of $I-Au_{60}$, $I_h-Au_{60}$, and $C_1-Au_{60}$ clusters. **a** Radial distribution functions for the three icosahedral shells depicted in Fig. 1c, e, g. The initial $I_h-Au_{60}$ structure (red) has a radius of ~7.1 Å, i.e., all 60 atoms lie precisely 0.71-nm from the center-of-mass. The final structure (black), $I-Au_{60}$, is nearly 5.9 Å in radius, or 1.18-nm in diameter. The $I-Au_{72}$ structure (green) exhibits two radii, of 6.03 and 6.86 Å, in the ratio 60:12, and reflects the two distinct sets of symmetry-equivalent sites. The surprisingly large contraction of the chiral icosahedral shell $I-Au_{60}$ (black) structure indicates unusually short (strong) bonds, and a high coordination number (5). **b** Comparison of bond lengths (nearest-neighbor distances) in the optimized $I-Au_{60}$ (black) and $I-Au_{72}$ (green) structures depicted in Fig. 1e, g.

(2.70 Å), blue in Fig. 1e, compose the 12 regular pentagons in the $I-Au_{60}$ structure, and map directly onto the Archimedean polyhedron in Fig. 1f. These distances are nearly 9% contracted from that of the bulk FCC gold. These same bonds become the longest ones (2.83 Å) in the $I-Au_{72}$ structures, the green peak at

farthest right. The next-shortest bonds (~2.74 Å in $I-Au_{60}$; 2.77 Å in $I-Au_{72}$) are the 30 radiating in a clockwise twist from each pentagon (these remain golden in Fig. 1e). Next are the bonds of the red triangles; these are longest in $I-Au_{60}$ (2.79 Å) and slightly shorter (2.76 Å) in $I-Au_{72}$. Finally, unique to the $I-Au_{72}$ structure are the links of the pentagons to each of the 12 icosahedral vertices; these are the shortest Au—Au distances (~2.75 Å) in the $I-Au_{72}$ structure. The geometric relations of $I-Au_{60}$ to $I-Au_{72}$ are stellation of the former, or truncation of the latter; the same holds true for multishell structures with an $I_h-Au_{55}$ core, i.e., $I-Au_{115}$ to $I-Au_{127}$. For the case of the ideal polyhedra of edge one, the rhombicosidodecahedron has ~4% larger radius and ~7% larger surface area than the snub dodecahedron. However, these geometrical principles do not predict the additional contraction depicted in Fig. 2a.

The transition from a rhombicosidodecahedron (3.4.5.4) to a snub dodecahedron ($3^4.5^*$) geometry is remarkably robust, and thorough density-functional theory (DFT) convergence tests were performed to investigate and analyze the atomic structure and structural stability of the 60-atom Au nanoshells. For a detailed description of the parameters and additional calculations performed with more sophisticated functionals, see Supplementary Methods. In summary, the calculations using the generalized-gradient approximation (GGA) exchange—correlation functional of Perdew-Burke-Ernzerhof (PBE)[21] are sufficiently accurate to support the results presented here. The present work reports the symmetrized coordinates found by the Symmol tool[22].

Profound chirality is an intrinsic feature that is quantified by the Hausdorff chirality measure (HCM)[17,23] or chiral index. The HCM of $C_1-Au_{60}$, $I_h-Au_{60}$, $I-Au_{60}$, and $I-Au_{72}$ are 0.009, 0.018, 0.092 and 0.079, i.e., 0.9%, 1.8%, 9.2% and 7.9% of diameter, respectively, as compared to the ideal $3^4.5^*$ (HCM = 10.1%). These values indicate that not only the structural transition to the $I$- symmetry is accompanied with an increase of the HCM index of chirality, but also that the $I-Au_{60}$ is more chiral than $I-Au_{72}$. Interestingly, the $I_h-Au_{60}$ shell displays weak chirality as it is shown by the non-zero HCM value.

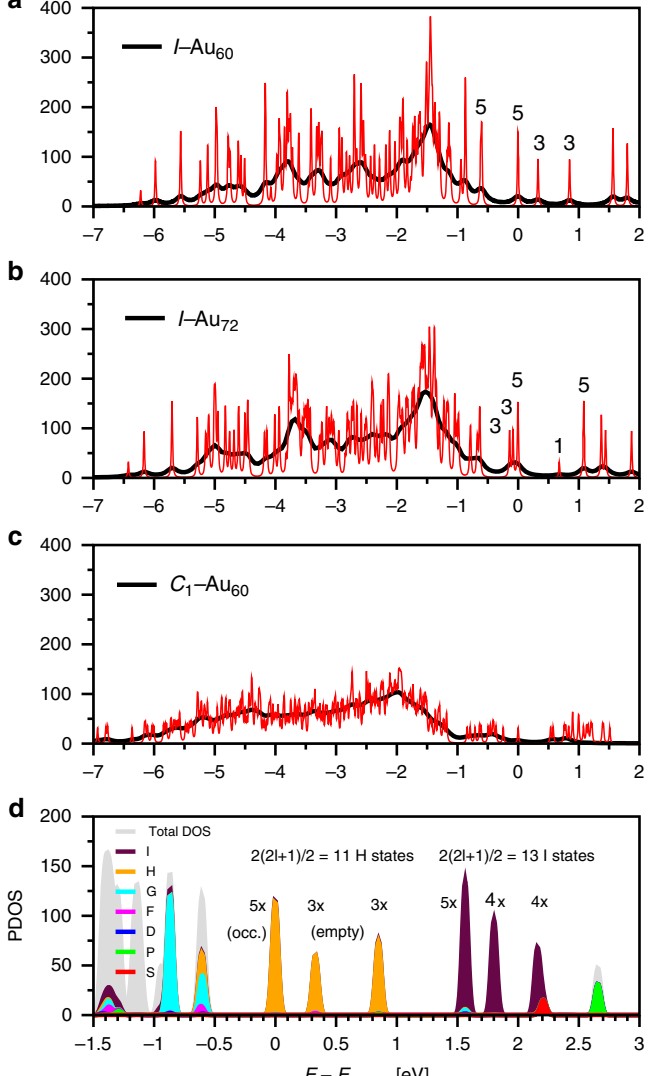

**Fig. 3** Electronic-structure comparison of $I-Au_{60}$, $I_h-Au_{60}$, and $C_1-Au_{60}$ clusters. **a–c** Electronic density of states (eDOS) of $I-Au_{60}$, $I-Au_{72}$, and $C_1-Au_{60}$ clusters. Degeneracies are indicated by the numbers. To highlight both the general shape of the eDOS and the discreteness of some parts of it, we draw the eDOS convoluted with Lorentzians of different width, viz., with $\Gamma = 0.1$ (black) and $0.01$ eV (red). The $I-Au_{60}$ is clearly more strongly peaked and shows more strongly separated, degenerate levels around the HOMO-LUMO gap than the $I-Au_{72}$. The $d$ band differs only little between these two structures. The eDOS of the two shell structures are very different from the much less peaked eDOS of the 3D $C_1-Au_{60}$ where hardly any degeneracies are found. **d** Projected density of states of $I-Au_{60}$ shell

The respective structure factors for $C_1-Au_{60}$, $I-Au_{60}$, and $I-Au_{72}$ are crucial for comparison with results from X-ray or electron scattering (diffraction) measurements. For the chiral-icosahedral shells, $I-Au_{60}$ and $I-Au_{72}$, the structure-factor I(s) patterns exhibit profound similarities. The shape-factor oscillations, in the small-s (low-angle) region, also reflect the overall greater size and symmetry of the chiral-icosahedral shells, see Supplementary Figure 4.

**Electronic structure of the $I$-symmetry $Au_{60}$ shell.** Figure 3a, b displays the highly discrete electronic density of states (eDOS) of the $I-Au_{60}$ and $I-Au_{72}$ clusters. The high degeneracies shown in the eDOS reflect the structures' $I$-symmetry. By contrast, the

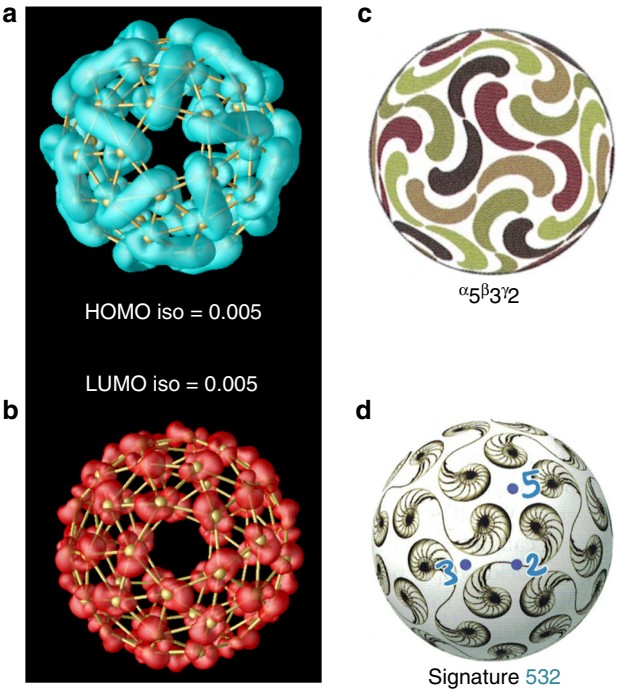

**Fig. 4** HOMO and LUMO states of $I-Au_{60}$ and chiral-icosahedral symmetry diagrams. At left (**a**, **b**) are representations of the electron densities of **a** the fivefold degenerate occupied (HOMO) states and **b** the three (3) unoccupied LUMOs, viewed along a $C_5$ symmetry-axis (Cf. Figure 3a for the energies of these levels.). In particular, the HOMO shows the unique characteristics of the frontier states of the chiral-icosahedral $I$—$Au_{60}$ cage. In **a**, one observes 30 ribbon-like 'bonds' that link and cup the Au-sites (These bonds are bisected by the 30 $C_2$-axes, are golden in Fig. 1e, bisecting the white rhombi in Fig. 1f, and are 2.74 Å in 1a.). In **b**, the densities (60 red cups) indicate non-bonding (or antibonding) character, but these states are unoccupied. To the right are the mathematical representations of chiral-icosahedral ($I$) symmetry, Copyright 2008 From The Symmetries of Things by Conway, Burgiel and Goodman-Strauss. Reproduced by permission of Taylor and Francis Group, LLC, a division of Informa plc.[24]. In **c**, the color-matched pairs of apostrophes (30 pairs in all) align with the blue ribbons (or bonds) in **a**, as do the tenuous threads in **d**. The mathematical notation (532) is identical (synonymous) in meaning to the symbol ($I$), corresponding to the cohabitation of fivefold, threefold, and twofold axes of rotation (marked in blue, in **d**), but no planes of symmetry

eDOS of the $C_1-Au_{60}$ displays a smeared lineshape characteristic of the low-symmetry 3D Au clusters in this size range[18], see Fig. 3c. The calculated HOMO-LUMO gap of the $I-Au_{72}$ cluster is 0.66 eV, which compares well with the one reported in reference[4] obtained with a similar DFT methodology (0.72 eV). The corresponding HOMO-LUMO gap for the $I-Au_{60}$ is smaller (0.32 eV), see below. Figure 3d shows the projected density of states (PDOS) of the $I-Au_{60}$ shell. Figure 4a, b shows the HOMO (5) and the LUMO (3) states of the $I-Au_{60}$ shell as seen along one of the fivefold rotation axes for the indicated values of the isosurfaces. This figure allows one to visually inspect the profound chirality of the frontier electron densities—separately for the set of (5) HOMOs and the set of (3) LUMOs. For the reader's convenience, we have added for comparison the diagrams used by mathematicians to analyze chiral icosahedral symmetry[24], see Fig. 4c, d.

The standard interpretation, based on a spherical shell model[25], indicates that since each Au(0) atom is in the $5d^{10}6s^1$ configuration, the neutral N-atom cluster has 10N electrons

localized in the filled $5d^{10}$ band, and N electrons in the delocalized 6s-band. In fact, the 5d-band is broad, ~4.0 eV in full width, extending from −5.0 to −1.0 eV in Fig. 3. But for the N electrons in the conduction band, the high (nearly spherical) symmetry generates an electronic structure corresponding to a sequential filling of the $1S^2$, $1P^6$, $1D^{10}$, $1F^{14}$, $1G^{18}$ ($L = 0–4$) shells, whereas the 11 frontier levels from the 1H-shell ($L = 5$) generate the (5) HOMOs and (3+3) LUMOs for $I$−$Au_{60}$.

Figure 3d shows how these groups of pure states split according to icosahedral symmetry. The 5−3−3 ordering is that of $I_h$−$C_{60}$ as well as for the icosahedral $Ag_{152}(SR)_{60}$[26]. At higher energy, the 13 I states ($L = 6$) also split as 5−4−(3 + 1). For $I$−$Au_{72}$, the 1H-shell is completely occupied and the ordering is reversed (3−3−5), but the splitting nearly vanishes, as consistent with the identified spherical aromaticity[4]. Neutral $I$−$Au_{60}$ and $I_h$−$C_{60}$ present only subshell closing (10/22); for this reason, both have high electron affinity and gain stability as anions, see below and Supplementary Note 1. Note that the magic numbers of shell-closings are not the same as in the case of a compact quasi-spherical clusters. This is expected as the radial part of the potential is entirely different in the $Au_{60}$ shell. To investigate further the stability of the $I$−$Au_{60}$, we have calculated the free energy of the $I$−$Au_{60}$ icosahedral shell and 3D $C_1$−$Au_{60}$ isomers, see Supplementary Figure 5.

**Vibrational spectrum and singular rigidity**. To investigate the structural stability of the $I$−$Au_{60}$ shell, the vibrational frequency spectra was obtained and compared with the $C_1$−$Au_{60}$ and $I$−$Au_{72}$ clusters, see Fig. 5. These results confirm that the three cluster-structures correspond to true local minima of their potential energy surface, since all frequencies calculated are real and significantly positive, see Supplementary Table 1 and Supplementary Data 5. The frequency range, 18–185 cm⁻¹, is the characteristic one of metallic gold clusters[27]. In particular, the range of the frequency spectrum of the $I$−$Au_{72}$ cluster agrees well with that calculated with a similar methodology in reference[4].

Consistent with their high $I$-symmetry configurations, the vibrational density of states (vDOS) lineshape of the cage-like 60- and 72-atom clusters shows a more structured profile as compared with that of the 3D $C_1$−$Au_{60}$ cluster. In fact, a symmetry analysis of the frequency spectrum establishes that, for example, the $I$−$Au_{60}$ cluster has 15 fivefold (H) and three (3) non-degenerate (A) Raman-active vibrational modes, plus seven (7) threefold ($T_{1u}$) IR-active ones. It should be noticed that this characterization of the vibrational modes for the $I$−$Au_{60}$ cluster differs from that displayed by the carbon fullerene $I_h$−$C_{60}$[25,28], which only has eight (8) fivefold ($H_g$) and two (2) non-degenerate ($A_g$) Raman-active vibrational modes, plus four (4) threefold ($T_{1u}$) IR-active ones[29]. This difference can be considered as another signature characterizing the $I$-symmetry in metal clusters.

Singular rigidity is evident not only from the larger contraction of the Au−Au bonds in $I$−$Au_{60}$ with respect to $I$−$Au_{72}$, but also in the larger values obtained for the highest frequencies of its vibrational spectrum. The highest five frequencies of $I$−$Au_{60}$ are in the range ~188 cm⁻¹, whereas in $I$−$Au_{72}$, this range is located around ~181 cm⁻¹. These higher frequency vibrations are related to tangential atomic motions corresponding to combinations of symmetrical and asymmetrical stretching modes, indicating a larger stiffness of the Au−Au bonding in $I$−$Au_{60}$, see Supplementary Figure 6 and Supplementary Movies 2−6 for animation of these modes. To further support the existence of high energy barriers which would keep the $I$−$Au_{60}$ in a metastable state, we performed molecular dynamics simulations. Our results indicate that the $I$−$Au_{60}$ nanocage remains stable up

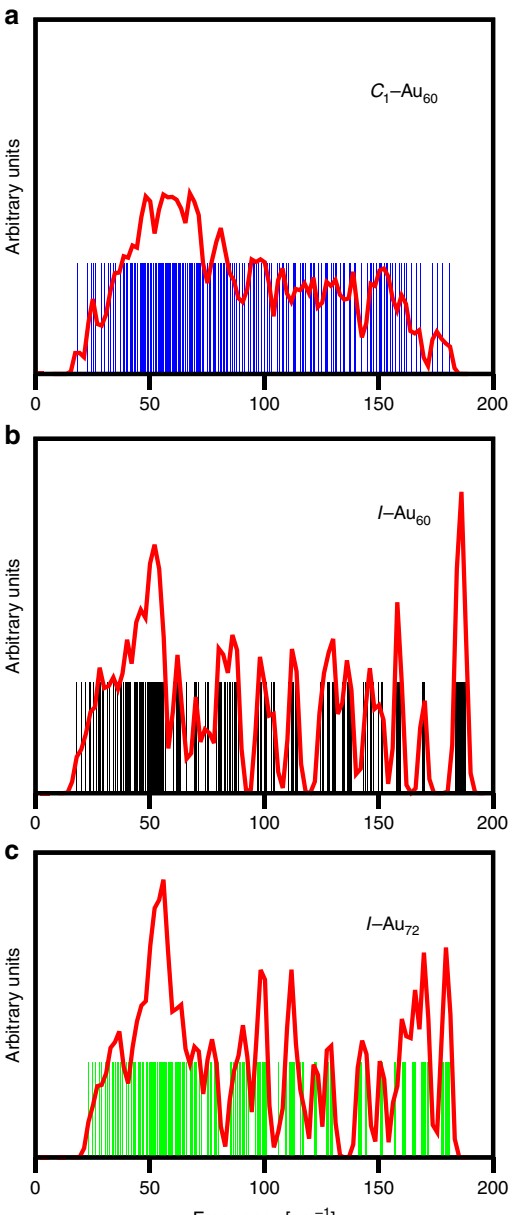

**Fig. 5** Comparison of the spectrum of vibrational frequencies, or density of states (vDOS) of $C_1$−$Au_{60}$, $I$−$Au_{60}$, and $I$−$Au_{72}$ clusters. **a** For the compact structure (see Fig. 1h), the vDOS resembles that of the incipient 3D lattice of bulk FCC gold, i.e., acoustic (to ~80 cm⁻¹) and optical (>80 cm⁻¹) phonons. **b, c** The spectra of the chiral-icosahedral shells (Fig. 1e, g) differ qualitatively from the spectra of the compact structure, showing a quasi-continuum of shape-deformation modes in the 20–60 cm⁻¹ range, and then a highly structured pattern of higher frequency modes reflective of the quasi-2D lattice (including its topological defects). The extreme simplicity (regularity or sparseness) in the case of $I$−$Au_{60}$, in **b**, reflects the 60-fold equivalence of all atomic sites. The large solitary spike, at the highest frequency ~190 cm⁻¹, provides a quantitative measure of its enhanced rigidity

to about 200 K, see Supplementary Figures 7, 8 and Supplementary Movies 7–11 for details and animations.

**Optical spectra of chiral quantized metallic cages**. We systematically compare the optical response of the $I$−$Au_{72}$ and $I$−$Au_{60}$ shells, by discussing the differences in their electronic absorption spectra. Figure 6 shows the calculated real-time time-dependent

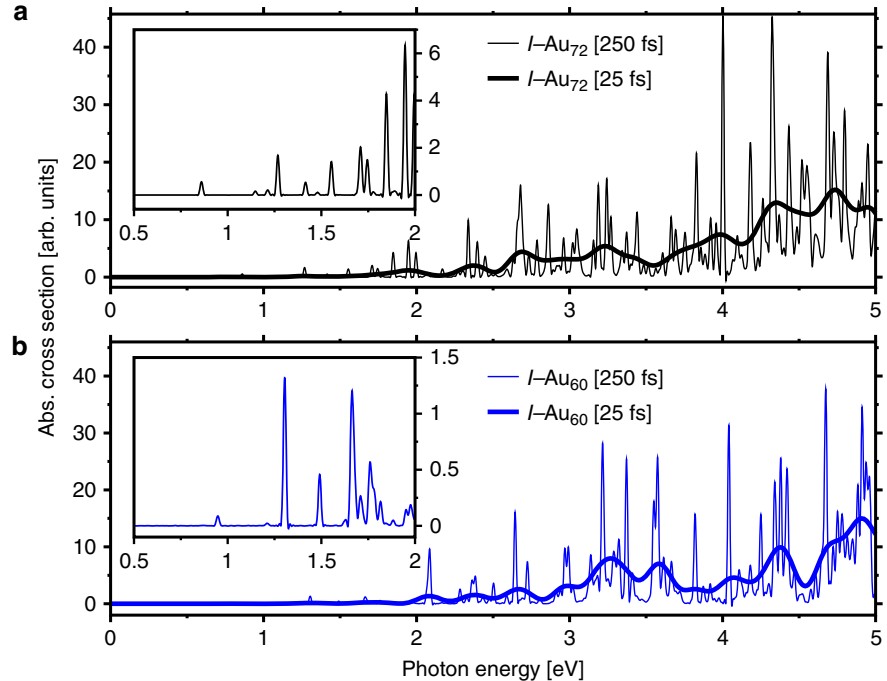

**Fig. 6** Optical absorption spectra of the chiral-icosahedral shells. The optical absorption spectra are computed from the time evolution of the electron density, over periods of 25 fs (low resolution, bold lines) and 250 fs (high resolution, fine lines) for **a** the $I-Au_{60}$ and **b** the $I-Au_{72}$. The insets expand the low-energy (0.5–2.0 eV) range, in order to highlight the distinct patterns of the $I-Au_{60}$ vs. $I-Au_{72}$ systems

DFT (TD-DFT) spectra for the indicated time evolution (25 fs and 250 fs). To ensure that all differences between the spectra are solely due to the geometry differences, we have used the same numerical setup and parameters, as described in the Supplementary Methods. The richly structured optical absorption spectra show that the chiral-icosahedral shells, $I-Au_{72}$ and $I-Au_{60}$, share many similarities, but differ especially in the 2.0 eV and 3.1–3.7 eV regions. In this range, the strongest peak at 3.25 eV of $I-Au_{60}$ consists of primarily two peaks and a few more weaker transitions. In contrast, the achiral-icosahedral shells $I_h-Au_{42}$ and $I_h-Au_{92}$ appear less regular in the spacings of major features[4,30]. One also observes an unusually discrete spectrum in the 1.0–3.0 eV region. The TD-DFT spectra for longer evolution time show that the broad features in the more strongly broadened spectrum are made up of several sharp transitions in all cases except for the peak at about 2.1 eV, which is due to one strong transition only.

A highly discrete spectrum in the red and near-infrared regions is of enormous practical significance. They may be used to identify the presence of $I-Au_{60}$ in mixture. A strong, discrete absorption or well-defined emission bands have many applications like optical sensing, optical microscopy, and labeling, cf. the case of the buckyball $C_{60}$. We also calculated the optical spectra for charge states (0, ±1, ±2) and compare them to the neutral structure, see Supplementary Figure 9. These results provide an estimate as to how much the optical spectra depend on charge states and thus are useful for comparison to experimental studies. Essentially, the charge-dependence of the spectra in the visible region is rather weak, showing a gradual, uniform shift of the spectral features toward higher energies. To explore the nature of the absorption features visible at different energies in the spectrum, we present the distinct modes of induced-density oscillations corresponding to the absorption at energies 1.31, 2.10, and 3.25 eV (see Supplementary Figure 10 and Supplementary Note 2). We demonstrate the interplay between the overall dipole-mode oscillation of the delocalized metal electrons and the

more strongly localized $5d$ electrons. In particular, the induced density highlights the chiral nature of the $I-Au_{60}$ shell.

Finally, in comparison with $I-Au_{72}$, the spectrum in the visible range seems to be dominated by fewer, stronger transitions. We conclude that these are effects of high symmetry and of the profound energy-level quantization depicted in Fig. 3 and electric-dipole selection rules governing transitions among well-defined angular-momentum quantum states. For a more detailed discussion of pervasive quantum information in the spectra of protected $Au_{144}$ cluster see reference[15].

**Discussion**

The emergence and stability of $I-Au_{60}$ warrant a comprehensive examination of its structural principles and mechanical characteristics, an exploration of its predicted electronic (bonding) and optical response properties, and some consideration of the prospects for its eventual detection and production by experimental means.

The perfect $I$-symmetry outcome deserves an explanation. The $I-Au_{60}$ shell structure predicted herein has not been experimentally characterized so far, and it may appear an unlikely candidate for experimental detection and physical isolation or chemical synthesis, in view of the greater cohesion of filled (compact) structures and the presumed reactivity of its exterior surface. However, its special properties render it ideal for such production: (i) structurally, it is $I$–spherical, of a convenient (1.178-nm) diameter, with minimum surface area, and no site is more vulnerable to attack than any other; (ii) mechanically, its short and strong bonds, all tangential to the surface of the sphere, render it unusually stiff or resistant to deformation; (iii) electronically, the prevalent $\{6s-5d_{z^2}\}$ hybridization (relativity derived) favors 2D-directional bonding[4,19,20]; (iv) (electro)chemically, its high electronegativity and set of electronic-shell closures $\{0, 6^-, 12^-\}$ reduce its reactivity and increase the stability of its anionic compounds. Its dimension (size) and rigidity (i–ii) suggest a templating approach.

For example, we have found that a $C_{60}$[28] fits almost perfectly within its interior void, i.e., to generate nested cages: $I_h-C_{60}$@ $I-Au_{60}$. There are still very few investigations of the interactions of gold shells with $C_{60}$[28]. In particular, the structural stability of gold fullerene-like structures covering the $I_h-C_{60}$ molecule was investigated for the icosahedral shell series of $Au_{32}$, $Au_{60}$, $Au_{66}$, and $Au_{92}$ by Batista et al.[31], wherein the most stable structure identified was $I_h-C_{60}$@ $I_h-Au_{92}$. Instead, we find the short and strong Au−Au bonding permits all 60 Au atoms to fit compactly around the $I_h-C_{60}$, versus only 32 for the alkaline-earth metals (Ca, Sr, Ba) explored by Martin[9]. The gas aggregation of 60 Au atoms upon a $C_{60}$ core is thus predicted, although this method would likely also produce the $I_h-C_{60}$@ $I-Au_{72}$.

The combination of 12-fold symmetry, electronic orbital structure, and high electro-negativity suggest several schemes involving electropositive metal-based counter-ions. For example, it could be obtained in the gas phase, or as a cluster beam, in the form $(A^+)_{12}-Au_{60}^{[12^-]}$ where the 12 $A^+$ ions are located in the 12 pentagons, (see eDOS and PDOS in Fig. 3). Besides the obvious choice of $A$ = alkali ions, interesting complexes might be formed with 12 mono-positive $MnCp^+$ or $FeCp^+$ that confer magnetic properties ($Mn^{2+}/Fe^{2+}$ ions) plus protection from the exterior Cp-groups. [$Cp^-$ = cyclopentadienide anions, $C_5H_5^-$][32]. Instead of metal-based counter-cations, one could also employ 12 exterior molecular cations, e.g., of the quaternary ammonium type, $R_4N^+$, which could also be chiral agents allowing for enantiomer selection. Combinations of templating and counter-ions may also be contemplated, e.g., the $I_h-C_{60}$@ $I-Au_{60}$ may benefit from 12 $M^{2+}$ divalent cations, given that each shell separately prefers the maximal [$12^-$] configuration. This list does not preclude other strategies; for example the special (local) electronic structure implied by (iii) suggests selective stabilization via the high affinity to Lewis bases, e.g., $NH_3$ or homologs. Xia et al. explore extensively synthesis techniques for many similar icosahedral nanostructures in the 10–250 nm size range, which may offer alternatives routes toward synthesizing this irreducible $I-Au_{60}$ structure[33].

Accordingly, exploratory efforts via DFT-PBE calculations show that the $I-Au_{60}$ shell can be stabilized further through various cores, namely $I_h-C_{60}$@$I-Au_{60}$, $I_h-C_{60}$@$I-Au_{72}$, $I_h-Au_{32}$@$I-Au_{60}^{2+}$ bilayer structures and $I-Au_{60}(MgCp)_{12}$. We find that the $I-Au_{60}$ shell structure is quite stable, for it can be obtained in several robust forms. In all cases, the $I$-symmetry is preserved, and the shell is slightly deformed only for the case of $I_h-C_{60}$@$I-Au_{60}$ system, see Supplementary Figure 11 and Supplementary Note 3.

Herein we have described how the achiral $I_h-Au_{60}$ shell, an ~(3.4.5.4)-polyhedron, spontaneously transforms into a quite distinct $I-Au_{60}$ structure, an ~($3^4.5^*$)-polyhedron, in which any small fluctuation determines which one of the two enantiomeric forms results. For this reason, the process is termed a chiral (icosahedral) symmetry breaking, although the chirality is generated, rather than destroyed in the transition. (The symmetry-elements destroyed are rather the unique inversion-center and manifold reflection planes.) From a purely geometrical standpoint, this transition may be regarded as simple: assuming fixed edge-lengths (bond distances), the resulting structure is not only more compact and spherical, but the coordination number at each site also increases. The actual physico-chemical situation is more complex, as three distinct edge-lengths are involved, and one had no a priori guarantee that these dimensions would remain fixed. However, the results presented here suggest that the edge-lengths decrease, resulting in a further contraction of the spherical shell.

From a topological standpoint, a more general and satisfying result emerges: an optimal infinite 2D net is 6-coordinated, but cannot be connected into a closed net, of any size, according to the Euler's Theorem, without introducing 'defects', specifically twelve (12) 5-coordinated sites. Placing these 12 defects maximally distant from each other naturally situates them on the vertices of a regular icosahedron. If the neighbors of these $12 \times 5$-coordinated sites are further constrained to participate in at most one defect, then it follows that $12 \times 5 = 60$ is the smallest number of such sites. If the defect sites are all occupied, one finds a minimum number of 72, but this comes at the expense of (i) convexity (Fig. 1g) and (ii) complexity (not all 72 sites are equivalent). If they remain vacant, then one obtains directly the special stability of the icosahedral network of precisely 60 5-coordinated sites, all of which are symmetry equivalent. But this description necessarily implies the $I_h$-symmetry-broken $I-Au_{60}$ structure. An analogous situation relates to the 3-coordinated 2D infinite and finite-closed nets, as popularized by graphene and fullerene structures, the latter of which obey a rule dictating the non-adjacency of pentagons.

These geometrical, mechanical, and topological arguments do not necessarily guarantee the high rigidity and the electronic stability (in neutral and 6- or 12-charge-states); these features emerged only in the course of high-level quantum electronic-structure investigations and depend ultimately on the high relativistic contraction of the gold-atom cores, which allows the $6s$ electrons to penetrate deeply and hybridize maximally with the symmetry-appropriate $5d$ orbital, which enhances the stability and rigidity of Au−Au bonding in reduced dimensionalities. Anderson has explained the existence of a generalized rigidity as a consequence of symmetry breaking in the context of phase transitions[34]. Whereas this argument is applied most fully in the thermodynamic limit (infinite-size systems), we believe it is appropriate here, since the high rigidity of the chiral $I-Au_{60}$ cluster results from the symmetry breaking of the $I_h-Au_{60}$ shell. Further analogies and connections between the symmetry breaking and its consequences at the nanoscale, and those existing in macroscopic condensed matter, remain to be investigated.

More broadly, exceptional fascination and broad interest have attended the discovery of novel structures and the aesthetics of their symmetry. The history of the carbon-cage $I_h-C_{60}$ 'fullerene' cluster-molecule illustrates this; fascination with its novel structure and bonding long preceded its impact in diverse fields. This structure is achiral; only much larger, so far unobserved, fullerenes may have $I$-symmetry[35]. The possible circumstance of profound chirality combined in a cluster with extremely high symmetry, the equivalence of all 60 Au-atom sites (the order of the chiral-icosahedral group is 60), manifests in the electronic structure where it matters most for applications, i.e., the frontier orbitals (HOMOs & LUMOs) involved in most chemical reactions, including enantiomeric selection and chiral catalysis. Critically, such insights have not previously appeared in the physical and materials-science literatures. To our knowledge, the icosahedral phases of the quasi-crystals (Nobel Prize 2012) make no mention of the possibility of chiral-icosahedral symmetry.

The findings reported herein should have an immediate and profound impact upon those working actively on several fronts: First, it should stimulate researchers to review their results extant, to check whether a hollow golden shell, of ca. 1.2 nm diameter, accounts for their observations in microscopy, crystallography, spectroscopy, mass spectrometry, etc. Second, it will stimulate new attempts to generate, detect, identify, and collect $I-Au_{60}$ species, in various media. Third, broad interest should be provoked by our findings indicating the need to elucidate the specific role that relativity plays in elemental gold's relativistic contraction and extreme $5d-6s$ hybridization, i.e., the bond length contraction (associated here specifically with enhanced rigidity) and the unusual competitiveness of quasi-2D structures vs. conventional

3D ones, especially when they are charged negatively. Fourth, it should attract attention and effort to understand this particularly fascinating class of symmetry breaking, and enhanced rigidity, that occurs in molecule-scale (<2-nm) structures of exceptionally high symmetry, yet generates a profound chirality that will be manifested in many ways, e.g., a strong chiroptical response (circular dichroism, optical rotation) in specific regions of the electromagnetic spectrum. Finally, quantum wave functions of chiral-icosahedral ($I$) symmetry have hardly been considered[35], e.g., in certain higher carbon-fullerene cages well beyond the size of any yet isolated.

Other recent reports strongly support the broader applicability of our approach and relevance of our findings: Trombach et al.[41] developed topological arguments overlapping ours as a general route to identifying exceptionally stable hollow gold cages; they also reported that the infinite hexagonal 2D net attains ~ 82% of the cohesion of bulk 3D gold, supporting one argument we employed for the experimental accessibility of the larger finite gold cages. In addition, chiral icosahedral symmetry has recently been positively identified in two very recent crystallographic reports[42] on compounds of the ubiquitous $Au_{144}$-cluster (two distinct compounds, with both enantiomeric forms identified in each case), which establishes our starting premise (Fig. 1a), as well as our concluding statement that an adequate conceptual understanding of chiral-icosahedral electronic structure and bonding will become of broad interest in chemistry, solid-state physics and nanoscience fields. Our results thus provide important theoretical evidence of the existence of a remarkably stable shell structure with much potential for future applications. This work should motivate the experimental synthesis of this $Au_{60}$ shell and $Au_{60}$ shell-based nanostructures. It remains to be investigated further analogies and connections between the symmetry breaking and its consequences at the nanoscale and those existing in macroscopic condensed matter.

## Methods

**Computational details**. The DFT relaxations were performed with the SIESTA code[36,37]. The GGA-PBE functional[21] and Norm-conserving Troullier-Martins (TM) pseudopotentials (PPs)[38] with scalar relativistic correction were used. The vibrational spectra of the $C_1$–$Au_{60}$, $I$–$Au_{60}$, and $I$–$Au_{72}$ clusters were calculated within the same DFT-GGA-PBE approximation, basis set and pseudopotentials using the VIBRA utility of the SIESTA code[36,37]. Absorption spectra were calculated using TD-DFT as implemented in the real-space code octopus[39,40] with TM PPs and the GGA-PBE approximation. A full description of the computational details and parameters can be found in the Supplementary Information file. It includes a description of all the convergence tests developed on different $Au_{60}$ shells, details on the molecular dynamics simulations, as well as additional calculations performed using more sophisticated functionals.

**Data availability**. The data that support the findings of this study are available from the corresponding author upon request.

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

## Acknowledgements

R.L.W. acknowledges support from Welch Foundation Grant #AX-1857. I.L.G. acknowledges support from DGAPA-UNAM under Project PAPIIT IN108817, CONACYT-México under Project 285821, and DGTIC-UNAM Supercomputing Center under Project LANCAD-UNAM-DGTIC-049. H.C.W. acknowledges support from the French Research Agency (Agence National de Recherche, ANR) in the frame of the project "FIT SPRINGS", ANR-14-CE08-0009 and HPC resources from GENCI-IDRIS (Grant 2016-096829). X.L.L. acknowledges previous funding from NSF-DMR-1103730 and NSF-PREM DMR-0934218. This work received computational support from the Laboratory of Computational Nanotechnology and UTSA Research Computing Support Group through the HPC clusters Antares3 and SHAMU, respectively, as well as the Texas Advanced Computing Center (TACC) at the University of Texas at Austin. R.S.-R. acknowledges support from the A*MIDEX Grant (No. ANR-11-IDEX-0001-02) funded by the French Government "Investissements d'Avenir" program.

## Author contributions

R.L.W., I.L.G. and X.L.-L. designed the study. S.M.M., H.-Ch.W., R.S.-R. and X.L.-L. performed the electronic structure and stability analysis. H.-Ch.W. and R.S.-R. obtained the optical response properties. I.L.G. and J.J.P. obtained the vibrational spectra. All authors participated in the analysis of the theoretical results, and in the writing of the manuscript.

## Additional information

**Competing interests:** The authors declare no competing interests.

