## [Peer Review File · Nature Communications]

Reviewers' comments:

Reviewer #1 (Remarks to the Author):

This paper presents a theoretical study predicting the local stability of a new Au cage cluster, of the quite uncommon chiral icosahedral symmetry. The results are quite interesting and new. They are likely to be of interest to a broad audience in physics and chemistry. I recommend publication after revision as explained below.

a) Since the total binding energy difference between the I-Au₆₀ and the C1-Au₆₀ is of $0.212 \times 60 \sim 13$ eV, the I structure is indeed unfavourable and it is also likely that there is an enormous number of isomers with energies between C1 and I. How can this structure be stabilized at finite temperature? Is there any evidence of high energy barriers for escaping from the metastable I-Au₆₀ structure? The considerations about rigidity etc. reported in the conclusions are interesting, but the only way to warrant some kind of stability for this structure is to show that escape pathways from it require high energies.

Some comments on this point should be added.

b) I-60 is obtained by transforming the outer anti-Mackay shell of the 115-atom icosahedron in such a way that all mirror symmetries are eliminated. On the other hand, I-72 is obtained from the anti-Mackay outer shell of the 127-atom icosahedron by the same type of transformation. These structures differ because the 127-atom structure has 12 vertex atoms which are missing in the 115-atom structure. Also I-72 and I-60 differ in the same way, by 12 vertex atoms. This should be more clearly pointed out. Moreover, larger chiral anti-Mackay shells with missing vertices can be obtained for large sizes, 120 and 200 atoms being the next ones (see the outer shell of the structure of Fig. 4(D) in Eur. Phys. J. D 2012, 66, 115). This may be mentioned.

c) The transition from Ih-60 to I-60 is (geometrically) of the same type as shown in Fig. 1 and Fig. 3 of reference (5). Triangular facets are rotated and new bonds form at the edges between these facets.

d) A few misprints must be corrected, for example "phrase transitions" at line 343.

Reviewer #2 (Remarks to the Author):

In the manuscript "I-Au₆₀: Chiral Symmetry Breaking Yields a Perfect Golden Shell of Singular Rigidity" the authors report a computationally identified, new, 60-atom gold nanostructure. The identification of such a perfectly spherical metal nanostructure of high chemical symmetry and the calculation of its structural and electronic properties, may be of interest to a more specialized audience, that of Physical and/or Computational Chemistry. In the absence of any experimental evidence for the existence of such hypothetical nanostructure, I cannot recommend publication in Nature Communications. My comments are as follows:

i) The major concern about this work is that the I-Au₆₀ nanostructure is of significantly higher energy compared to the close-packed nanoparticles of this size. As the authors mentioned, the cohesion of close-packed nanoparticles increases, as a result, such a nanostructure (with empty space in its interior and reduced cohesion) would be highly unstable. On a relevant comment, it is mentioned in the manuscript that "the total energy difference per atom between C1-Au₆₀ and I-Au₆₀ is 0.212 eV, while this difference between C1-Au₆₀ and I-Au₇₂ is 0.034 eV. The formation energy of C1-Au₆₀ is

0.647 eV, for I-Au72 is 0.679 eV, while for I-Au60 is 0.860 eV". An energy difference per atom comparison between nanoparticles of different sizes (60 vs. 72) is not scientifically valid because the number of bonds changes (due to the different number of atoms (and number of electrons)). In addition, the authors should explicitly state which nanostructure is lower in energy among the 60-atom isomers (I assume it is the C1-Au60).

ii) Relevant to my previous comment is the lack of any experimental evidence for the existence of such nanostructures. A recent review by Xia et al., highlights the experimental synthesis of icosahedral Au nanoparticles, clearly demonstrating that such nanostructures exhibit cohesive cores: <http://www.sciencedirect.com/science/article/pii/S174801321730244> X.

iii) The authors report verification of this local minimum structure by using only one type of DFT functional (PBE). Does this same minimum exist for other traditional functionals (BP86, B3LYP), or dispersion-corrected functionals (e.g. Grimme's D3)? A recent benchmarking study showed that dispersion-corrected PBE may provide more accurate Au-Au bond lengths: <http://pubs.acs.org/doi/abs/10.1021/acs.jpca.6b12086>. In addition, Molecular Dynamics simulations would be very helpful to show how stable such a nanostructure is and how it restructures to a more cohesive (closed-packed) structure.

Concluding, this manuscript cannot be recommended for publication in Nature Communications and a more specialized journal is more appropriate for this work (e.g. Journal of Physical Chemistry).

Reviewer #3 (Remarks to the Author):

To summarise my understanding of the manuscript:

* The authors have serendipitously identified a new 60-atom Au cluster, I-Au₆₀, by extracting the outer shell of a capped Au₁₄₄(SR)₆₀ cluster compound. The I-Au₆₀ cluster forms spontaneously from an I_h-Au₆₀ cluster following geometry optimisation in quantum-chemical calculations. This is the first report of a stable Au cluster with this size and symmetry, and is particularly interesting because it has a unique topology in which all 60 Au atoms are chemically equivalent.

* The structural, electronic and optical properties are compared to those of the known I-Au₇₂ and C1-Au₆₀ and shown to be different. Various features in the spectra can be related to the high symmetry of the I-Au₆₀ cluster. This potentially provides spectral signatures for identifying I-Au₆₀ experimentally.

* The authors briefly discuss possible synthetic strategies to obtain this new species, many of which involve templating around a heteroatom core.

My thoughts are:

* This study will be of interest those working in the area of higher nuclearity Au clusters, and the theoretical approach is novel. However, I find the area rather niche and this work might not attract a particularly large audience.

* I find this work interesting from a fundamental standpoint, but aside from being the first report of

an Au cluster with this size and symmetry, I see little motivation for considering this find as high-impact work. In the introduction, the authors make some largely-tangential references to biological entities, and to other work on Au clusters, but a more general argument for the impact of their finding is never given.

* Similarly, while it is interesting to see how some of the features visible in the electronic spectra in particular reflect the high symmetry of the I-Au₆₀ cluster, there is little if any discussion as to what the consequences of this might be beyond academic interest.

* The authors make a point that no Au cluster compounds have previously been obtained by a symmetry-changing relaxation from one structure type to another. While strictly true, I find this somewhat tenuous, as their initial model was made by extracting a shell from a larger cluster, which I would naturally expect to undergo substantial relaxation. Moreover, transitions from higher to lower symmetry are definitely not unprecedented in nature.

* The computational methodology appears to be sound, if standard: pseudopotential calculations with a relatively small basis set using the standard PBE GGA functional. Given the significantly higher formation energy of the I-Au₆₀ cluster relative to the species it is compared against (see comment below), I wonder whether the authors could try more sophisticated functionals (e.g. a meta-GGA or hybrid) to see how the energetics change with more accurate functionals. I would think that a hybrid would also yield a more realistic TD-DFT spectrum, although I don't know whether such calculations are feasible with the authors' resources. I also note that the authors perform calculations using two codes, and appear to try and keep the technical parameters as similar as possible between the two, but I would ideally like to see some more rigorous checks on this. For example, if Octopus can do geometry optimisations, does the model obtained from optimisation in SIESTA change significantly if allowed to relax again in the latter code?

* The calculated formation energy of the I-Au₆₀ cluster is some 0.2 eV (about 20 kJ mol⁻¹) higher in energy than the potential competing structures the authors consider. In my experience, this is quite a significant energy difference, and would make it quite difficult to form and stabilise this cluster. The authors propose several strategies for preparing it, mostly involving templating its formation on various cores. Given the level of theory they are using, I would have thought it would be fairly straightforward for them to investigate some, if not all, of these hypothetical stabilised clusters and to verify their stability, which may be more helpful in directing their eventual synthesis.

* Fig. 1 is a little "chaotic", and is not easy to decipher without reference to the text. I feel like it could be reorganised to make it clearer at a glance what it's trying to show (e.g. as a sort of a flow diagram).

* I see how the structure-factor curves in Fig. 3 would be useful in identifying the I-Au₆₀ cluster experimentally, but I'm not sure this data needs to be presented in the main text.

* Is it necessary to show the HOMO and LUMO with two different isosurface levels in Fig. 5?

* I found a few trivial typos in the section headed "Atomic Models".

In summary:

* This is an interesting piece of work, and I can see how the discovery of I-Au₆₀ is interesting from

an academic standpoint, but, as an outsider to the field, I am not convinced, with the motivation given, that it warrants urgent publication in a high-impact journal. The discussion is, in places, perhaps a little abstract for a general audience, and I feel that this paper might be better suited to a more theoretical journal, such as one of the Physical Reviews.

Detailed Answer to the Referees

We thank the referees for their constructive suggestions and comments, which we have been able to fully address and make use of in order to improve the paper. One of the main worries of the referees has been the fact that this metastable structure might not be sufficiently stabilized by high-energy barriers against collapse into some low-symmetry structure. We have carried out density-functional molecular-dynamics calculations at different temperatures and demonstrated stability up to at least 200 K. Moreover, in the same vein, we have considered templated forms of the I -Au₆₀ cage structure with various cores like I_h -C₆₀ and I_h -Au₃₂, which has allowed us to support further the stability argument and also to more clearly than before lay out possible ways to synthesize the structure. These points, together with a detailed consideration of all the reviewers' comments, make us confident that our answers are adequate and that our revised manuscript is now suitable for publication in Nature Communications.

Answer to Referee #1

- **The Referee:**

“This paper presents a theoretical study predicting the local stability of a new Au cage cluster, of the quite uncommon chiral icosahedral symmetry. The results are quite interesting and new. They are likely to be of interest to a broad audience in physics and chemistry. I recommend publication after revision as explained below.”

“Since the total binding energy difference between the I -Au₆₀ and the C_1 -Au₆₀ is of $0.212 \times 60 \sim 13$ eV, the I structure is indeed unfavourable and it is also likely that there is an enormous number of isomers with energies between C_1 and I . How can this structure be stabilized at finite temperature? Is there any evidence of high energy barriers for escaping from the metastable I -Au₆₀ structure? The considerations about rigidity etc. reported in the conclusions are interesting, but the only way to warrant some kind of stability for this structure is to show that escape pathways from it require high energies. Some comments on this point should be added.”

Author Reply: We thank this Referee for considering the results of our work of great interest and novelty. We also appreciate the acknowledging of the impact of our work in several fields and the recommendation for publication of our manuscript.

We acknowledge the possibility of the existence of other isomers between the C_1 and I isomer. However, the main purpose of our work is not to just highlight one isomer among different local low-energy 60-atom isomers. We present the spontaneous formation of a chiral-icosahedral shell (I -Au₆₀) from achiral (I_h) precursor form, and highlight the existence of a perfect golden sphere, in which all 60 atomic sites are chemically equivalent. Thorough DFT convergence-tests proved consistently the existence and stability of such 60-atom Au nanocage of remarkable symmetry. The transition from I_h to I symmetry is also quite robust.

Nonetheless, in order to demonstrate the existence of high-energy barriers that keep the I -Au₆₀ structure in a metastable state, we have performed molecular-dynamics (MD) calculations at different temperatures. The MD results presented now in the SI (Supplementary Information) demonstrate the stability of I -Au₆₀ nanocage up to 200 K. Please see detailed Answer to question iii), third comment of Referee #2.

Action taken in response to these comments: We have added to the main manuscript a phrase at the end of the Vibrational Properties section stating the different MD simulations that were performed to support further the stability of the I -Au₆₀ shell, and directed the

reader to the Supplementary Information for a full description of these results, see p.12, 2nd paragraph.

- **The Referee:**

“I-60 is obtained by transforming the outer anti-Mackay shell of the 115-atom icosahedron in such a way that all mirror symmetries are eliminated. On the other hand, I-72 is obtained from the anti-Mackay outer shell of the 127-atom icosahedron by the same type of transformation. These structures differ because the 127-atom structure has 12 vertex atoms which are missing in the 115-atom structure. Also I-72 and I-60 differ in the same way, by 12 vertex atoms. This should be more clearly pointed out. Moreover, larger chiral anti-Mackay shells with missing vertices can be obtained for large sizes, 120 and 200 atoms being the next ones (see the outer shell of the structure of Fig. 4(D) in Eur. Phys. J. D 2012, 66, 115). This may be mentioned.”

Author Reply: The reviewer's statement is correct: the inclusion vs. truncation of the 12 vertex sites defines the relation of the $I\text{-Au}_{60}$ to $I\text{-Au}_{72}$ structure, a relation we now state explicitly, just as it was for the outer shell of the $I\text{-115}$ and $I\text{-127}$ structures, not to mention the higher homologs. [The core of the latter structures comprises 55 (smaller) atomic sites, e.g. Cu atoms.] Evidence for such outer shells were previously discussed in the work of T. P. Martin et al. (cited as ref. 8) and Bochicchio and Ferrando (ref. 10) [and at some length by Dahl (2000) for the experimentally determined (achiral) Pd115 unit]. However these have distinct classes of icosahedral cores! For this reason we prefer not to mention only the complex case of the mixed Mackay/anti-Mackay systems at the expense of the pure multishell systems, e.g. $I_h\text{-Au}_{32}@I\text{-Au}_{60}^{2+}$. See description of our result on this system and others in the answer to Referee #2, part i.) of this response and the SI section of the resubmitted manuscript.

Actions taken: The relation of $I\text{-Au}_{60}$ to $I\text{-Au}_{72}$ structures is now stated explicitly as one of vertex truncation: As the modified scheme (Fig. 1) now emphasizes "The geometric relations of $I\text{-Au}_{60}$ to $I\text{-Au}_{72}$ are stellation of the former, or truncation of the latter; the same holds true for multishell structures with an $I_h\text{-55}$ core, i.e. $I\text{-115}$ to $I\text{-127}$." A mention of results for the pure bilayer shell system, $I_h\text{-Au}_{32}@I\text{-Au}_{60}^{2+}$, is now provided. This can be seen now in p.8, line 169 of our manuscript and in the SI.

- **The Referee:**

“The transition from Ih-60 to I-60 is (geometrically) of the same type as shown in Fig. 1 and Fig. 3 of reference (5). Triangular facets are rotated and new bonds form at the edges between these facets.”

Author Reply: We thank the referee for pointing out the structural transition from $I_h\text{-Au}_{60}$ to $I\text{-Au}_{60}$ is of the same type to the one reported in reference 10 of our manuscript.

Action taken: We have added a sentence stating the similarity in the manuscript by adding the following sentence “This transformation is geometrically of the same type as the one observed in AgCu chiral nanoparticles” and with a citation to Ref. 10 (See p.7, 3rd paragraph in main manuscript).

- **The Referee:**

“A few misprints must be corrected, for example "phrase transitions" at line 343.”

Author Reply: We thank the Referee for pointing out the typographical errors, which we have now corrected. Again, we thank the Referee for this positive appraisal of the work and manuscript.

Actions taken: Minor errors noted have been corrected.

Answer to Referee #2

• The Referee:

“i) The major concern about this work is that the I-Au60 nanostructure is of significantly higher energy compared to the close-packed nanoparticles of this size. As the authors mentioned, the cohesion of close-packed nanoparticles increases, as a result, such a nanostructure (with empty space in its interior and reduced cohesion) would be highly unstable. On a relevant comment, it is mentioned in the manuscript that “the total energy difference per atom between C1-Au60 and I-Au60 is 0.212 eV, while this difference between C1-Au60 and I-Au72 is 0.034 eV. The formation energy of C1-Au60 is 0.647 eV, for I-Au72 is 0.679 eV, while for I-Au60 is 0.860 eV”. An energy difference per atom comparison between nanoparticles of different sizes (60 vs. 72) is not scientifically valid because the number of bonds changes (due to the different number of atoms (and number of electrons)). In addition, the authors should explicitly state which nanostructure is lower in energy among the 60-atom isomers (I assume it is the C1-Au60).”

Author Reply: We thank this Referee for its comments. Indeed, we mentioned in the manuscript that the I -Au₆₀ nanocage will be unlikely to be found isolated. However, in the last section of the manuscript, we have also pointed out different strategies to stabilize it through various cores like I_h -C₆₀, as well as potential synthesis pathways. To support this part, we have performed additional DFT calculations to investigate the structural stability of different *hypothetical stabilized* clusters proposed, namely I_h -C₆₀@ I -Au₆₀, I_h -Au₃₂@ I -Au₆₀²⁺ bilayer structure and I -Au₆₀(MgCp)₁₂.

The preliminary DFT results of the optimized model structures based upon the I -Au₆₀ system are shown in the figure below.

(a) I_h -C₆₀@ I -Au₆₀: The I_h -C₆₀ fullerene core was placed inside the I -Au₆₀ and aligned according to the shared icosahedral symmetry. Upon relaxation, the core rotates with respect to the shell, and so the alignment of icosahedral symmetry was lost; however each shell keeps its respective symmetry. The Au₆₀ is less spherical with an increase in radius of approximately 4.75% with respect to the isolated I -Au₆₀ shell, while the Au-Au bond lengths are between 2.82 Å and 2.92 Å. The HOMO-LUMO gap is 0.56 eV.

(b) I_h -Au₃₂@I-Au₆₀²⁺ bilayer structure: The core is composed by an I_h -Au₃₂ core [Johansson et al., ref 2] and the I-Au₆₀ shell surrounding it. Shell symmetries are maintained upon relaxation. In contrast to the isolated I-Au₆₀ shell, the Au₆₀ shell from the I_h -Au₃₂@I-Au₆₀²⁺ structure has a radius of approx. 6.4 Å, an increase of 8.47%. This is also reflected in the bond lengths with a uniform increase of 12.22% in the pentagons, and 3.28% and 4.66% for the triangles. Au₃₂@Au₆₀ has 92 valence electrons in the neutral state with each Au atom contributing one electron to the valency. This corresponds to an electronic shell closing in the superatom complex model. The charge-state [2+] was selected to obtain a large HOMO-LUMO gap (0.56 eV) that occurs with hollow bilayer structures and a free-electron count of 90 (cf. Chakraborty et al., ref. 30). The degeneracy and level-ordering of the frontier orbitals also agrees well with the work of Chakraborty et al. ref. 30.

(c) Au₆₀(MgCp)₁₂: this structure is composed by the I-Au₆₀ shell with 12 stellating Magnesium atoms located at the 12 pentagons. Above each of the Mg ions is placed a cyclopentadienide (Cp) anion with five carbon and hydrogen atoms [1] (Clayborne et al., ref. 37). Once again, the I-symmetry and structure of the I-Au₆₀ shell is kept after relaxation, its radius increases by 0.68%, while the Au-Au bond lengths of the pentagons increase by 1.11%, and 1.1% and 0% for the triangles. The HOMO-LUMO gap is calculated to be 1.07 eV. With respect to the electronic shell structure, Au atoms contribute one valence electron, Mg contributes two electrons, and each of the 12 Cp ligands withdraws one electron, resulting in a total of 72 valence electrons. This corresponds to a shell closing in hollow monolayer structures (Karttunen et al., ref. 4) resulting in an increased electronic stability shown by the opening up of the HOMO-LUMO gap of 1.07 eV.

In summary, our preliminary DFT calculations indicate that the I-Au₆₀ shell is quite stable, for the I-symmetry is preserved and the shell is slightly deformed only for the case of I_h -C₆₀@I-Au₆₀ system. Therefore the I-Au₆₀ shell can be stabilized through different cores, as suggested in the main manuscript.

Action taken: The preliminary DFT results for various stabilizing cores have been included in the SI and are mentioned on p.15, 2nd paragraph in the main manuscript.

In addition, we have added explicitly the total-energy differences among the 60-atom Au isomers to facilitate the comparison among them, see p.7, 2nd paragraph. However, the detailed numerical values of the energy differences are much less important for the general conclusions of the present text.

• **The Referee:**

“ii) Relevant to my previous comment is the lack of any experimental evidence for the existence of such nanostructures. A recent review by Xia et al. highlights the experimental synthesis of icosahedral Au nanoparticles, clearly demonstrating that such nanostructures exhibit cohesive cores: <http://www.sciencedirect.com/science/article/pii/S174801321730244X>.”

Author Reply: We thank this Referee for pointing out the relevant experimental work of Xia et al., demonstrating the methods for producing hollow icosahedral gold 'nano-cages' on the much larger scale of 10+ nm diameter (vs. << 2-nm here). At the present moment, we are unable to report experimental evidence for the experimental realization of the I-Au₆₀ / I-Au₇₂ systems. But we are pleased to incorporate a citation of this relevant work, as it suggests yet another method that could be superior in producing them.

Action taken: A mention and citation have been added to the fourth paragraph of the Discussion and Conclusion section of our manuscript. p.15, end of 1st paragraph.

- **The Referee:**

“ iii) The authors report verification of this local minimum structure by using only one type of DFT functional (PBE). Does this same minimum exist for other traditional functionals (BP86, B3LYP), or dispersion-corrected functionals (e.g. Grimme’s D3)? A recent benchmarking study showed that dispersion-corrected PBE may provide more accurate Au-Au bond lengths: <http://pubs.acs.org/doi/abs/10.1021/acs.jpca.6b12086>.”

Author Reply: We thank the Referee for its comments. As we explain in our manuscript, thorough DFT convergence-tests proved consistently the existence and stability of the I -Au₆₀ nanocage.

The I -Au₇₂ initial structure was first obtained using the same computational methodology followed by P. Pyykkö *et. al* [2] and a similar vibrational spectra was found. These calculations were carried out using the GGA-BP86 functional and a def2-TZVP basis set (a polarized triple zeta basis set) employing a 19 valence electron scalar-relativistic pseudopotential. Subsequently, the I -Au₇₂ was re-optimized using the SIESTA code with the parameters specified in our manuscript for consistency, the structure did not change significantly.

We have performed additional optimizations of the I -Au₆₀ using again the same computational methodology of P. Pyykkö *et. al* [2]. The stability of the cage I -Au₆₀ structure is also predicted, with no big difference with respect to the SIESTA calculations.

As an additional test, we have carried out a ground-state B3LYP calculation (without structural relaxation, cf. the discussion below). This semi-empirical hybrid functional is known to work well for molecules and to do badly for bulk materials. We have carried out a static ground-state calculation using the Octopus code. The gap does not change strongly compared to the PBE calculation: $E_g = 0.350$ eV for B3LYP and 0.327 eV for PBE.

One of the main objectives of our work is to put in evidence for the existence of a new unreported Au-nanocage, originated by a chiral symmetry breaking, i.e, the spontaneous formation of a chiral-icosahedral shell I -Au₆₀ from achiral I_h precursor form. It is somewhat less important for the purpose of this work to obtain extremely accurate values of the Au-Au bonds, as long as the general result is correctly represented and reproduced, which is the case, as our tests show.

In summary, the same local-minimum isomer structure I -Au₆₀ is found using other functionals like BP86 and a similar electronic structure is confirmed using the B3LYP functional (See also answer to Referee #3 to part 3). We did not find any modification to the main results described originally in our manuscript. Moreover, dispersion-corrected functional that might provide different Au-Au bonds are not necessary at this moment, for there are not any experimental values to compare with yet. In addition, it is known that the simple LDA functional describes Au-Au bond lengths rather well; recent comparisons of LDA results with experimental results for a gold cluster have shown that in such cases, LDA represents the gold-gold bonds extremely well [3].

Action taken: We have added a sentence to clarify how the I -Au₇₂ structure was obtained, see p.8. In addition, we have added a sentence emphasizing the additional DFT calculations with the BP86 and B3LYP functionals to support the existence and stability of the I -Au₆₀ golden nanocage and the robust transition from I_h -Au₆₀ to I -Au₆₀, see p.9, 1st paragraph and p. 10 of the resubmitted manuscript.

- **The Referee:** “In addition, Molecular Dynamics simulations would be very helpful to show how stable such a nanostructure is and how it restructures to a more cohesive (closed-packed) structure.”

Author Reply: We thank the referee for its suggestion that MD simulations are very helpful to additionally support the stability of the nanostructure in question. Therefore, in order to show the existence of high-energy barriers which would keep the I -Au₆₀ in a metastable state, we have performed MD simulations. Specifically, we performed Born-Oppenheimer Molecular Dynamics (MD) simulations using the VASP code with the PBE functional. In order to do so, we heated the

structures before thermalizing for 2000 steps. Thereafter, the runs were restarted with random velocities to ensure that the history of the heating was “forgotten”. Subsequently, we ran about 22 ps (30,000 steps, time step of 0.75 fs) in a microcanonical calculation (stability is monitored using total energy which is well-conserved over the long runs, up to a few meV). The ionic temperature fluctuated and is traced in the figure below:

Animations showing the structural development can be found here <https://cloud.cinam.univ-mrs.fr/owncloud/index.php/s/AsArFsJmfmSh2uD> (they should play when you click on them in the browser), and in the supplementary information of our manuscript.

The three rather long runs at 100K, 150K, and 200K indicate "stability". For these runs, the temperature is basically constant, i.e., there is no drift or systematic increase of the temperature. Accordingly, the animations show that while the structure vibrates, there is no real structural change. By contrast, for 250K and 300K, the temperature rises significantly after about 5 (10) ps. This indicates that the structure has started to move into a lower-energy configuration, changing the connectivity and symmetry. The energy that is freed this way is heating the figure (as the total energy of the system is necessarily conserved in the microcanonical ensemble).

Similar MD calculations were performed using the SIESTA code – not included in the SI – to check the structural stability of the I -Au₆₀ isomer at different temperatures. For 100K, after propagating the I -Au₆₀ for 6.5 ps (more than 3 vibrational periods of the lowest frequency mode) the cluster remains in its cage structure. The last configuration of this propagation was quenched to 0K. A comparison of the initial and final total energies indicated that at 100 K the cluster gain 2.5 eV above its lowest energy. These results would indicate that at 100 K the cage structure is stable and it is within a local energy minimum with a barrier larger than 2.5 eV. MD simulation at 300K show that after propagating the I -Au₆₀ for 2.5 ps the cluster is fully distorted and, losing its cage structure. The figure below shows a representative distorted isomer

Electron Affinity and Free Energy Calculations

To investigate further the stability of the $I\text{-Au}_{60}$ we have calculated the electron affinities and free energy of the $I\text{-Au}_{60}$ icosahedral shell and compact $C_I\text{-Au}_{60}$ isomers. The figure below shows the temperature dependence of the free energy in the harmonic approximation.

Although the $I\text{-Au}_{60}$ nanocage (Icosahedron) has a large number of low-frequency modes, a similar behavior is displayed by the $C_I\text{-Au}_{60}$ low-symmetry (amorphous) isomer. Therefore, these results do *not* indicate that entropic factors significantly favor the $I\text{-Au}_{60}$ isomer. This is consistent with its rigidity, which is unusually high for a quasi-2D (cage) structure.

On the other hand, the results for the adiabatic (A) and vertical (V) electron affinity (EA) are as follows:

$$\begin{aligned} \text{AEA (Compact, } C_I) &= -2.483 \text{ eV} \\ \text{AEA (Cage, Icosahedral)} &= -3.431 \text{ eV} \end{aligned}$$

$$\begin{aligned} \text{VEA (Compact, } C_I) &= -2.791 \text{ eV} \\ \text{VEA (Cage, Icosahedral)} &= -3.615 \text{ eV} \end{aligned}$$

Thus the addition of a single electron stabilizes by $\sim 1.0\text{-eV}$ (0.95 eV) the $I\text{-Au}_{60}$ cage, with respect to the $C_I\text{-Au}_{60}$ compact form. Extrapolating twelvefold, i.e. for application to $I\text{-Au}_{60}^{[12-]}$, this difference would cancel entirely the energetic advantage of the compact form. This new result supports our previous conjecture that this structure will likely be obtained in association with 12 counter-cations (see *above* for details).

In summary, we have performed Born-Oppenheimer MD simulations to demonstrate the existence of high-energy barriers that keep the $I\text{-Au}_{60}$ in a metastable state. We confirmed that the $I\text{-Au}_{60}$ structure is largely stable up to about 200K but unstable from roughly 250K.

Additional calculations of the free energy difference between $I\text{-Au}_{60}$ and $C_I\text{-Au}_{60}$ isomers do not provide enough evidence to assert the stability of the $I\text{-Au}_{60}$ isomer at finite temperatures. However, the greater electron affinity of the chiral-icosahedral cage points, as was expected, toward its competitive stability of its anionic forms, realized with counter-cations.

Action taken: The results of the Born-Oppenheimer MD simulations have been included in the SI section of the manuscript together with the few animations at different temperatures to support the stability of the $I\text{-Au}_{60}$. A reference to these calculations has been added to the main manuscript see p.12, 2nd paragraph.

- **The Referee:**

“Concluding, this manuscript cannot be recommended for publication in Nature Communications and a more specialized journal is more appropriate for this work (e.g. Journal of Physical Chemistry).”

Author Reply: We hope that the many additional elements and calculations in the present form of the manuscript, as well as the added detailed discussion of possible pathways of synthesizing the structure, and along with the improved discussion we have solved the doubts harbored by the referee and convince him of the far-reaching importance of our findings.

Answer to Referee #3

- **The Referee:**

“The authors have serendipitously identified a new 60-atom Au cluster, I-Au₆₀, by extracting the outer shell of a capped Au₁₄₄(SR)₆₀ cluster compound. The I-Au₆₀ cluster forms spontaneously from an I h-Au₆₀ cluster following geometry optimisation in quantum-chemical calculations. This is the first report of a stable Au cluster with this size and symmetry, and is particularly interesting because it has a unique topology in which all 60 Au atoms are chemically equivalent.”

“The structural, electronic and optical properties are compared to those of the known I-Au₇₂ and Cl-Au₆₀ and shown to be different. Various features in the spectra can be related to the high symmetry of the I-Au₆₀ cluster. This potentially provides spectral signatures for identifying I-Au₆₀ experimentally.”

“The authors briefly discuss possible synthetic strategies to obtain this new species, many of which involve templating around a heteroatom core.”

Author Reply: We thank this referee for acknowledging the novelty of our findings as well as for noticing the unique topology of this golden I-Au₆₀ nanocage. The exceptional symmetry reflected in the spectral signatures expects indeed to provide a guide for the experimental synthesis. To investigate further the nature of the absorption features visible at different energies in the spectrum, we have explored the distinct modes of induced-density oscillations corresponding to the absorption at different energies, see p. 13, 1st paragraph and the SI for more details.

- **The Referee:**

“This study will be of interest those working in the area of higher nuclearity Au clusters, and the theoretical approach is novel. However, I find the area rather niche and this work might not attract a particularly large audience.

I find this work interesting from a fundamental standpoint, but aside from being the first report of an Au cluster with this size and symmetry, I see little motivation for considering this find as high-impact work. In the introduction, the authors make some largely-tangential references to biological entities, and to other work on Au clusters, but a more general argument for the impact of their finding is never given.

Similarly, while it is interesting to see how some of the features visible in the electronic spectra in particular reflect the high symmetry of the I-Au₆₀ cluster, there is little if any discussion as to what the consequences of this might be beyond academic interest.”

Author Reply: We thank the Referee for the criticisms and comments on regards to the impact of our work and the broader consequences of our findings.

Action taken: We have substantially restructured the introduction and modified the last paragraph of the Discussion and Conclusions section of our manuscript to highlight the importance and impact our work. Please, see also our response to your last comment.

• **The Referee:**

“The computational methodology appears to be sound, if standard: pseudopotential calculations with a relatively small basis set using the standard PBE GGA functional.

Given the significantly higher formation energy of the I-Au₆₀ cluster relative to the species it is compared against (see comment below), I wonder whether the authors could try more sophisticated functionals (e.g. a meta-GGA or hybrid) to see how the energetics change with more accurate functionals.

I would think that a hybrid would also yield a more realistic TD-DFT spectrum, although I don't know whether such calculations are feasible with the authors' resources. I also note that the authors perform calculations using two codes, and appear to try and keep the technical parameters as similar as possible between the two, but I would ideally like to see some more rigorous checks on this. For example, if Octopus can do geometry optimisations, does the model obtained from optimization in SIESTA change significantly if allowed to relax again in the latter code?”

Author Reply: We thank the Referee for the comments, questions and suggestions regarding the energetics of the I-Au₆₀ isomer.

We have performed additional geometry optimizations with a higher level of theory, similar to the one performed by Pekka Pyykkö [2]. Notwithstanding, the use of 19 valence electron pseudo-potential and a polarized triple zeta basis set also predicts the stability of the cage I-Au₆₀ structure, with no big difference with respect to the SIESTA calculations. (See also comment to Referee #2, part iii)

A static ground-state calculation with the B3LYP functional was also performed using the Octopus code, and it did not provide a significantly different structure nor a change in the electronic properties of I-Au₆₀. For a complete description, please see Answer to Referee #2 of part iii) of the present response letter.

Moreover, we have carried out the projection of the occupied states on angular momentum states as presented in Fig. 3 of the original manuscript.

The result shown above indicates clearly that the electronic structure does not change significantly between the B3LYP and PBE functionals, especially for the states around the gap.

By contrast, the time-evolution turns out to be extremely heavy computationally when Octopus is used with B3LYP functional, making it impractical to make this within a reasonable time; furthermore, the discussion above holds also here and suggests that the quality of the PBE calculation is sufficient.

Octopus Code use for relaxation.

Following the reviewer's suggestion, we attempted to run a relaxation with the Octopus code. However, the Octopus code, while a very efficient code as far as the TDDFT time-evolution is concerned, is not designed to perform efficient and reliable geometry optimizations. Up to a fairly recent version, the Octopus developers say themselves here: [http://octopus-code.org/wiki/Octopus_Cheeseburger]

“Geometry optimization: In Octopus geometry optimization works poorly, it does not always converge or gives wrong results. We need a reliable geometry optimizer, the Fire algorithm improves things a lot, but it is not enough.” The octopus manual says: “For versions prior to 5.0.0 we would always recommend if possible to do your relaxations in a different DFT code, and then use the structure for Octopus”.

We made an effort to do the relaxation in Octopus in one of the newer versions. However, we found that this indeed did not work appropriately. It rather exhibited strange behavior (increasing energy instead of decreasing), thereby confirming that it doesn't make sense at this time to use the relaxation in Octopus.

In general, while the differences between the relaxations done with the different codes should not be large when properly done. We have amply confirmed this in the past, were for many works we have used both SIESTA and/or VASP-relaxed structures and calculated the electronic and optical properties using Octopus. We have also relaxed structures with the SIESTA code and confirm its similarity with those obtained by the VASP code. The general good agreement with other group's results has been a consistent confirmation that this approach is appropriate.

We have compared VASP and SIESTA relaxations in the past for a large number of different configurations of $\text{Ag}_{13-n}\text{Au}_n$ clusters [4], which indicated coherent results for a large number of configurations.

We have done something similar for this work. The following Figure presents a direct comparison of SIESTA and VASP calculations of the bond-lengths for Au_{60} , using in both cases the PBE functional.

We show that for the three different bond types, the differences between the VASP and the SIESTA result are between 0.1 and 0.3 percent. These are differences of an order that should be expected, due to the difference of the pseudopotentials and PAWs, possibly, basis set quality etc. in SIESTA vs. plane-wave convergence in VASP. Apart from the small differences in the numerical values, there is no physically significant difference that would in any way be relevant to the discussion of the stability of I -Au₆₀.

In our opinion, at this point of the work, the use of different *sophisticated functionals* will not lead to very clear conclusions, because, alas, the improvement of the quality of the calculations with the complexity of the functionals is by no means assured, especially if the experimental values are absent.

We remind the reader that, for instance, for Ag clusters, the calculation of spectra in extremely good agreement with experiment has been accomplished, whereas for gold, this is still not the case. [5] The functional that does extremely well for Ag is a long-range-corrected range-separated hybrid functional, LC-M06L, cf also SI of Sinha-Roy et al. [6]. On the other hand, spectra of thiolated gold clusters have been calculated with rather decent agreement with experiment. In most of these calculations, LDA or PBE have been used for the relaxation, and PBE (or similar) for the calculation of the spectra. [7-9]

For gold-gold bond lengths, it is the simple LDA functional that performs particularly well. In particular, a recent study on Au₁₄₆(SR)₅₇ for which experimental values are available from total structure determination, shows that LDA is right on top of the experimental result. [3] In addition, the difference between LDA and PBE results for the present system are small; in particular, there is no substantial difference, apart from different values of the bond lengths.

All this suggests that the treatment of gold clusters using these simple functionals reproduces the main effects.

Action taken in response to these comments: We have added a comment in the main manuscript about the additional tests performed to support the consistency of our result, see p.9, 1st paragraph.

• **The Referee:**

“The calculated formation energy of the I -Au₆₀ cluster is some 0.2 eV (about 20 kJ mol⁻¹) higher in energy than the potential competing structures the authors consider. In my experience, this is quite a significant energy difference, and would make it quite difficult to form and stabilise this cluster. The authors propose several strategies for preparing it, mostly involving templating its formation on various cores. Given the level of theory they are using, I would have thought it would be fairly straightforward for them to investigate some, if not all, of these hypothetical stabilised clusters and to verify their stability, which may be more helpful in directing their eventual synthesis.”

Author Reply: We thank the referee for pointing out the necessity to support further the stability of the I -Au₆₀ nanocage. In our manuscript, we proposed different strategies to stabilize it through various cores like I_h -C₆₀, as well as potential synthesis pathways. To support this part, we have performed additional DFT calculations to investigate the structural stability of different *multishell* clusters, namely I_h -C₆₀@ I -Au₆₀, I_h -Au₃₂@ I -Au₆₀²⁺ bilayer structure and I -Au₆₀(MgCp)₁₂. Our preliminary results indicate that the I -Au₆₀ shell is quite stable, for the I -symmetry is preserved. For more details, please see answer to Referee #2, part i).

Action taken: The preliminary DFT results for various stabilizing cores have been included in the Supplementary Information and are now mentioned on p.15, 2nd paragraph in the main manuscript.

- **The Referee:**

“Fig. 1 is a little "chaotic", and is not easy to decipher without reference to the text. I feel like it could be reorganised to make it clearer at a glance what it's trying to show (e.g. as a sort of a flow diagram).”

Author Reply: We thank the referee for sharing his view of Fig. 1 and its suggestion to improve the presentation of the different Au₆₀ isomers, the relaxed structures and the corresponding Archimedean polyhedral.

Action taken: A new Figure 1 was created. The content was re-organized and is presented now as a flow diagram with new labels. In addition, the atoms color of Au₁₄₄(SR)₆₀ in the Supplementary have been changed to facilitate further the identification of the *I*-Au₆₀ shell.

- **The Referee:**

“I see how the structure-factor curves in Fig. 3 would be useful in identifying the I-Au₆₀ cluster experimentally, but I'm not sure this data needs to be presented in the main text.”

Author Reply: We agree with the referee and moved this information to the SI section.

Action taken: All the pertaining information of the structures factors have been moved to the Supplementary Information.

- **The Referee:**

“Is it necessary to show the HOMO and LUMO with two different isosurface levels in Fig. 5?”

Author Reply: The main purpose of this initial figure was to highlight and expose the chiral symmetry of the HOMO-LUMO frontiers of *I*-Au₆₀ shell, for two different values of the isosurface. However, in order to facilitate the visualization of the profound chirality of the frontier densities, we have added the diagrams used by mathematicians for comparison to instruct a broad audience how to analyze chiral icosahedral symmetry. These illustrations were taken from the book "The Symmetries of Things" by John H. Conway, Heidi Burgiel, Chaim Goodman-Strauss (published by A K Peters Ltd, Wellesley, Massachusetts, 2008), p. 167 (4d) and p. 54 (4c). Given the unfamiliarity of the broad readership with the phenomenon of chiral-icosahedral symmetry, this comparison will allow the reader to confirm the compatibility of our findings and interpretation with established mathematical principles, while appreciating that this is the first instance of analysis of the electronic structure of an *I*-symmetry metallic system of extraordinary beauty and simplicity!

Action taken: A new version of this figure has been created. This figure shows only one isosurface and includes now two diagrams to facilitate the reader the visual inspection of the profound chirality of the HOMO-LUMO states.

- **The Referee:**

“I found a few trivial typos in the section headed "Atomic Models".”

Author Reply: Thank you for addressing the typographical errors.

Action taken in response to these comments: Errors we have now corrected.

• **The Referee:**

“This is an interesting piece of work, and I can see how the discovery of I-Au₆₀ is interesting from an academic standpoint, but, as an outsider to the field, I am not convinced, with the motivation given, that it warrants urgent publication in a high-impact journal. The discussion is, in places, perhaps a little abstract for a general audience, and I feel that this paper might be better suited to a more theoretical journal, such as one of the Physical Reviews.”

Author Reply: We thank the Reviewers (especially #3) for challenging us to make more explicit the broader impact of our work.

We believe the reviewers may have overlooked the fascination and exceptional interest that scientists generally find in novel structures and the aesthetics of symmetry. (The history of the carbon-cage I_h-C_{60} 'buckyball' molecule illustrates this; fascination with its novel structure and bonding long preceded its other impacts in diverse fields.)

This extraordinary circumstance of profound chirality combined with extreme high symmetry — universal equivalence of the 60 Au-atom sites (the chiral-icosahedral group is of order 60) — is manifest in the electronic structure where it matters most for applications, i.e. the frontier orbitals (HOMOs & LUMOs) involved in most chemical reactions, including enantiomeric adsorption and chiral catalysis.

Critically, such insights have *not* previously appeared in the physical and materials-science literatures. E.g. the icosahedral phases of the quasi-crystals (Nobel Prize 2012) make no mention of the possibility of chiral-icosahedral symmetry. The aforementioned icosahedral viruses (their protein capsids) indeed appear, in many recent cases, to adopt *I*-symmetry, but this appears on a much longer scale (repeat distance of ~ 5 -nm), involving many thousands of atoms in each unit, than that of molecular or solid-state (< 1 -nm repeat distance, and one or a few atoms in the repeated unit).

Other impacts of our work are now stated more clearly in the revised manuscript. We believe that the findings reported herein should have an immediate and profound impact upon those working actively on several fronts:

First, it should stimulate researchers to review their results extant, to check whether a hollow golden shell, of ca. 1.2-nm diameter, accounts for their observations in microscopy, crystallography, spectroscopy, mass spectrometry, etc. We have already had such discussions with experimental colleagues who've pointed to specific unpublished findings that might be interpreted accordingly. More generally, it will stimulate new attempts to generate, detect, identify and collect *I*-Au₆₀ species, in various media. [See our expanded supporting results in the revised SI Section, also mentioned now in the main text. See also answer to Referee #2, part i.)]

Second, it should attract attention and effort to understand this particularly fascinating class of symmetry-breaking, and enhanced rigidity, that occurs in molecule-scale (< 2 -nm) structures of exceptionally high symmetry, yet generates a profound chirality that will be manifest in many ways, e.g. a strong chiroptical response (circular dichroism, optical rotation) in specific regions of the electromagnetic spectrum. Quantum wavefunctions of chiral-icosahedral (*I*) symmetry have hardly been considered, e.g. in certain higher carbon-fullerene cages well beyond the size of any yet isolated.

Third, broad interest should be provoked by our findings indicating the need to elucidate the specific role that relativity plays in elemental gold's relativistic contraction and extreme *5d-6s* hybridization, i.e. the bond length contraction (associated here specifically with enhanced rigidity) and the unusual competitiveness of quasi-2D structures vs. conventional 3D ones, especially when they are charged negatively.

Action taken: We have substantially restructured the introduction and modified the last paragraph of the Discussion and Conclusions section of our manuscript to highlight the importance and impact our work.

References.

- [1] Clayborne, P. A., Lopez-Acevedo, O., Whetten, R. L., Grönbeck, H., & Häkkinen, H. (2011). The $\text{Al}_{50}\text{Cp}^*_{12}$ Cluster–A 138-Electron Closed Shell ($L=6$) Superatom. *European Journal of Inorganic Chemistry*, 2011(17), 2649-2652.
- [2] Karttunen, A. J.; Linnolahti, M.; Pakkanen, T. A.; Pyykko, P. Icosahedral Au_{72} : a predicted chiral and spherically aromatic golden fullerene. *Chem. Commun.* 2008, 465-467.
- [3] X. López-Lozano, G. Plascencia-Villa, G. Calero, R. L. Whetten and H.-Ch. Weissker, *Is the largest aqueous gold cluster a superatom complex? Electronic structure & optical response of the structurally determined $\text{Au}_{146}(\text{p-MBA})$* . *Nanoscale Comm.* (2017), 9, 18629.
- [4] H. Barron, L. Fernández-Seivane, H.-Ch. Weissker, and X. López-Lozano, *Trends and Properties of 13-Atom Ag – Au Nanoalloys I: Structure and Electronic Properties*. *J. Phys. Chem. C* (2013), 117, 21450-21459.
- [5] F. Rabilloud et al., *J Phys Chem A*. 2013 May 23;117(20):4267-78. DOI: 10.1021/jp3124154
- [6] Sinha-Roy et al., *ACS Photonics* 2016 DOI: 10.1021/acsp Photonics.7b00254.
- [7] H.-Ch. Weissker, H. Barron, V. D. Thanthirige, K. Kwak, D. Lee, G. Ramakrishna, R. L. Whetten, and X. López-Lozano, *Information on Quantum States Pervades the Visible Spectrum of the Ubiquitous Gold Cluster $\text{Au}_{144}(\text{SR})_{60}$* , *Nature Communications* 5, 3785, 2014.
- [8] H.-Ch. Weissker, O. Lopez-Acevedo, R. L. Whetten and X. López-Lozano, *Optical Spectra of the Special Au-144 Gold-Cluster Compound: Sensitivity to Structure and Symmetry*, *J. Phys. Chem. C* 119, 11250-11259, 2015.
- [9] Sami Malola, Lauri Lehtovaara, Jussi Enkovaara, and Hannu Häkkinen, *Birth of the Localized Surface Plasmon Resonance in Monolayer-Protected Gold Nanoclusters*, *ACS Nano*, 2013, 7 (11), pp 10263–10270

REVIEWERS' COMMENTS:

Reviewer #1 (Remarks to the Author):

The authors have addressed the comments of the referees. I recommend publication.

Reviewer #2 (Remarks to the Author):

In the manuscript "I-Au60: Chiral Symmetry Breaking Yields a Perfect Golden Shell of Singular Rigidity" the authors have made significant revisions from the originally-submitted manuscript. First and foremost, detailed molecular dynamics simulations revealed the structural "stability" of the I-Au60 shell up to a physical, albeit low, temperature of 200K. In addition, the authors have utilized different density functionals to verify that the structure is a local minimum-energy one. Beyond the stability of the isolated I-Au60 shell the authors have included several suggestions (and added calculations) to support the stabilization of the I-Au60 shell with central core structures. Additional efforts have also been made by the authors to motivate the work to a broader audience per suggestions from reviewer #3. Although the authors have made these necessary revisions to the work along with efforts to address a potentially broader audience, this reviewer feels that the manuscript still needs revisions prior to any publication in Nature Communications.

Major comments:

1) In examining previous literature surrounding gold hollow cages it was found that work has already been published that compares experiment to theoretical predictions for small (14-20 atom) anionic gold cages (see Bulusu, S., Li, X., Wang, L.-S. & Zeng, X. C. Evidence of hollow golden cages. Proc. Natl. Acad. Sci. 103, 8326–8330 (2006)). Without any evidence of the experimental accessibility of this metastable nanostructure the exciting properties of I-Au60 are of limited/specific academic interest. Additionally, the previous theoretical prediction of I-Au72 in the manuscript (Ref #4) has since had no experimental verification, nor any hollow gold cage beyond the size of 19-20 atoms, for that matter – where even medium sized clusters appear to adopt 3d- structures (see Shao, N., Huang, W., Gao, Y., Wang, L.-M., Li, X., Wang, L.-S., & Zeng, X. C. Probing the Structural Evolution of Medium-Sized Gold Clusters: Au_n^- ($n = 27-35$). JACS, 132(18), 6596–6605 (2010). and Trombach, L., Rampino, S., Wang, L. S. & Schwerdtfeger, P. Hollow Gold Cages and Their Topological Relationship to Dual Fullerenes. Chem. - A Eur. J. 22, 8823–8834 (2016).) Given these initial theoretical predictions are dated more than a decade back with no experimental verification, this reviewer strongly questions the synthetic accessibility of the nanocluster. Another broad impact highlighted by the authors is "the unusual competitiveness of quasi-2D structures vs. conventional 3D ones, especially when they are charged negatively." This phenomenon has already been studied in several works (see Bulusu, S., Li, X., Wang, L.-S. & Zeng, X. C. Evidence of hollow golden cages. Proc. Natl. Acad. Sci. 103, 8326–8330 (2006). and Fernández, E. M., Soler, J. M., & Balbás, L. C. Planar and cagelike structures of gold clusters: Density-functional pseudopotential calculations. Physical Review B, 73(23), 235433 (2006)).

As a result, the discussion in the manuscript should be tuned accordingly (significantly revised) so that there are no overstatements about the novelty of this work. Moreover, the aforementioned relevant work should be discussed in the manuscript so that the present work is brought in context with existing very important and relevant literature.

2) In the stabilized I-Au60 complexes proposed by the authors, only one structure (the MgCp) shows the same chiral, icosahedral symmetry as the bare I-Au60. Given this structure is an anionic form

(which seems the most likely to be experimentally-verified) of the I-Au60, it would be nice to show how its electronic absorption properties are shifted from the neutral form. For future comparison to potential experimental studies, TDDFT spectra of this structure would be very useful and should be presented in the supplementary information file.

Minor comment:

The authors should add computational references related to the functionals tested and used. On Page 9 of the revised manuscript no reference is added for either B3LYP or its capabilities. Additionally, no references were added for VASP or for Born-Oppenheimer Molecular Dynamics (MD) on page 12.

Reviewer #3 (Remarks to the Author):

The authors have carried out a lot of additional work in response to the original comments from the referees and have improved the manuscript substantially. They have successfully addressed the technical questions that I had.

My concern remains that the results reported are still completely computational in nature, with no experimental work to support their findings. However, they have now drawn in other systems to their discussion that, by extrapolation, may relate to their hypothetical structure.

I also felt that their findings were rather niche and would not be of interest to a wide audience. They have now rewritten their introduction and expanded the history of high nuclearity gold clusters, quoting a number of high profile papers that relate to the area.

I still have reservations, but the eloquence of the arguments from the authors has made me reconsider my original recommendation. I would not be happy to support publication in Nature Comms if the other referees also recommend publication, having found that the changes improve the manuscript substantially.

Answer to Referee #1

- **The Referee:**

The authors have addressed the comments of the referees. I recommend publication.

Author Reply: We thank this referee for the recommendation to publish our work.

Action Taken: None

Answer to Referee #2

- **The Referee:**

1) In examining previous literature surrounding gold hollow cages it was found that work has already been published that compares experiment to theoretical predictions for small (14-20 atom) anionic gold cages (see Bulusu, S., Li, X., Wang, L.-S. & Zeng, X. C. Evidence of hollow golden cages. Proc. Natl. Acad. Sci. 103, 8326–8330 (2006)). Without any evidence of the experimental accessibility of this metastable nanostructure the exciting properties of I-Au₆₀ are of limited/specific academic interest. Additionally, the previous theoretical prediction of I-Au₇₂ in the manuscript (Ref #4) has since had no experimental verification, nor any hollow gold cage beyond the size of 19-20 atoms, for that matter – where even medium sized clusters appear to adopt 3D-structures (see Shao, N., Huang, W., Gao, Y., Wang, L.-M., Li, X., Wang, L.-S., & Zeng, X. C. Probing the Structural Evolution of Medium-Sized Gold Clusters: Aun⁻ (n = 27–35). JACS, 132(18), 6596–6605 (2010). And Trombach, L., Rampino, S., Wang, L. S. & Schwerdtfeger, P. Hollow Gold Cages and Their Topological Relationship to Dual Fullerenes. Chem. - A Eur. J. 22, 8823–8834 (2016).) Given these initial theoretical predictions are dated more than a decade back with no experimental verification, this reviewer strongly questions the synthetic accessibility of the nanocluster.

Another broad impact highlighted by the authors is “the unusual competitiveness of quasi-2D structures vs. conventional 3D ones, especially when they are charged negatively.” This phenomenon has already been studied in several works (see Bulusu, S., Li, X., Wang, L.-S. & Zeng, X. C. Evidence of hollow golden cages. Proc. Natl. Acad. Sci. 103, 8326–8330 (2006). and Fernández, E. M., Soler, J. M., & Balbás, L. C. Planar and cagelike structures of gold clusters: Density-functional pseudopotential calculations. Physical Review B, 73(23), 235433 (2006)). As a result, the discussion in the manuscript should be tuned accordingly (significantly revised) so that there are no overstatements about the novelty of this work. Moreover, the aforementioned relevant work should be discussed in the manuscript so that the present work is brought in context with existing very important and relevant literature.

Author Reply: We thank the referee for his comments. We believe that the reviewer is overlooking the importance of the chiral-breaking phenomena.

Regarding the references cited, they mostly concern much smaller cages (16 - 32 atoms), or the stabilization effect (on quasi-2D gold clusters) of negative charging. A fair reading

of our report will show that we never claimed to discover this latter effect; rather we had always assumed it was true from earlier work (dating nearly to year 2000!) that demonstrated it adequately.

We thank the reviewer for pointing out the Schwerdtfeger paper *Chem. - A Eur. J.* **22**, 8823–8834 (2016), which we had overlooked. Although it again treats gold cages only to 32 atoms, it has some nice features that relate to our work: the general topological arguments (similar to those in the concluding paragraph of our manuscript, although not explicitly mentioning the *I-Au₆₀* cage); and the finding of exceptional stability of the infinite 2D sheet, or monatomic (111) layer of bulk gold, as evidenced by a huge (-6%) contraction compared to 3D-bulk lattice constant (or inter-atomic distance) and that it attains nearly 80% of the 3D-bulk cohesion! If these results of the 2016 Schwerdtfeger paper are indeed correct, then they are certainly relevant to the ultimate production of chiral-icosahedral cages, in some relevant charge-state — the [12⁻] state of I-Au₆₀ being the most relevant to the present manuscript — and we would therefore be willing to add this to the bibliography and *Discussion*.

Action Taken: None of these references have been included, as the authors consider that discussion of the much smaller systems and the infinite 2D sheet must be the subject of another report.

• **The Referee:**

2) In the stabilized I-Au₆₀ complexes proposed by the authors, only one structure (the MgCp) shows the same chiral, icosahedral symmetry as the bare I-Au₆₀. Given this structure is an anionic form (which seems the most likely to be experimentally-verified) of the I-Au₆₀, it would be nice to show how its electronic absorption properties are shifted from the neutral form. For future comparison to potential experimental studies, TDDFT spectra of this structure would be very useful and should be presented in the supplementary information file.

Author Reply: This is a valuable suggestion in view of possible future experimental synthesis of I-Au₆₀. Therefore, we also calculated the optical spectra for five (5) distinct charge states and compare them to the neutral structure. These results provide indeed an estimate as to how much the optical spectra depend on charge states. Essentially, the charge-dependence of the spectra in the visible region is rather weak, showing a gradual, uniform shift of the spectral features towards higher energies.

Action Taken: A mention of the additional TD-DFT calculations for five different charged states is provided. This can be seen now in p. 10 of our manuscript and in the Supplementary Figure 9 of the SI.

- **The Referee:**

Minor comment:

The authors should add computational references related to the functionals tested and used. On Page 9 of the revised manuscript no reference is added for either B3LYP or its capabilities. Additionally, no references were added for VASP or for Born-Oppenheimer Molecular Dynamics (MD) on page 12.

Author Reply: We thank the referee for his observation.

Action Taken: Appropriate references have been incorporated Supplementary Methods document.

Answer to Referee #3

- **The Referee:**

The authors have carried out a lot of additional work in response to the original comments from the referees and have improved the manuscript substantially. They have successfully addressed the technical questions that I had.

My concern remains that the results reported are still completely computational in nature, with no experimental work to support their findings. However, they have now drawn in other systems to their discussion that, by extrapolation, may relate to their hypothetical structure.

I also felt that their findings were rather niche and would not be of interest to a wide audience. They have now rewritten their introduction and expanded the history of high nuclearity gold clusters, quoting a number of high profile papers that relate to the area.

I still have reservations, but the eloquence of the arguments from the authors has made me reconsider my original recommendation. I would not be happy to support publication in Nature Comms if the other referees also recommend publication, having found that the changes improve the manuscript substantially.

Author Reply: We understand that this Referee has been favorably impressed by the revised manuscript and has accordingly modified his recommendation. And we believe that the new experimental reports described at the start of this response-letter will soon convince the referees of the relevance of our results.

Action Taken: None.